# Increasing Prevalence of HIV-1 Transmitted Drug Resistance in Portugal: Implications for First Line Treatment Recommendations

**DOI:** 10.3390/v12111238

**Published:** 2020-10-30

**Authors:** Marta Pingarilho, Victor Pimentel, Isabel Diogo, Sandra Fernandes, Mafalda Miranda, Andrea Pineda-Pena, Pieter Libin, Kristof Theys, M. Rosário O. Martins, Anne-Mieke Vandamme, Ricardo Camacho, Perpétua Gomes, Ana Abecasis

**Affiliations:** 1Global Health and Tropical Medicine (GHTM), Instituto de Higiene e Medicina Tropical/Universidade Nova de Lisboa (IHMT/UNL), 1349–028 Lisbon, Portugal; victor.pimentel@ihmt.unl.pt (V.P.); a21000919@ihmt.unl.pt (M.M.); andreapinedap@gmail.com (A.P.-P.); mrfom@ihmt.unl.pt (M.R.O.M.); annemie.vandamme@kuleuven.be (A.-M.V.); ana.abecasis@ihmt.unl.pt (A.A.); 2Laboratório de Biologia Molecular (LMCBM, SPC, CHLO-HEM), 1349-019 Lisbon, Portugal; ifmadeira@chlo.min-saude.pt (I.D.); smfernandes@chlo.min-saude.pt (S.F.); gomes.perpetua@gmail.com (P.G.); 3Department of Microbiology and Immunology, KU Leuven, Clinical and Epidemiological Virology, Rega Institute for Medical Research, 3000 Leuven, Belgium; pieter.libin@vub.ac.be (P.L.); kristof.theys@kuleuven.be (K.T.); ricardojorge.camacho@kuleuven.be (R.C.); 4Artificial Intelligence Lab, Department of computer science, Vrije Universiteit Brussel, 1000 Brussels, Belgium; 5Interuniversity Institute of Biostatistics and statistical Bioinformatics, Data Science Institute, Hasselt University, 3500 Hasselt, Belgium; 6Centro de Investigação Interdisciplinar Egas Moniz (CiiEM), Instituto Superior de Ciências da Saúde Egas Moniz, 2829-511 Caparica, Portugal

**Keywords:** HIV-1, transmitted drug resistance, acquired drug resistance, Portugal

## Abstract

Introduction: Treatment for All recommendations have allowed access to antiretroviral (ARV) treatment for an increasing number of patients. This minimizes the transmission of infection but can potentiate the risk of transmitted (TDR) and acquired drug resistance (ADR). Objective: To study the trends of TDR and ADR in patients followed up in Portuguese hospitals between 2001 and 2017. Methods: In total, 11,911 patients of the Portuguese REGA database were included. TDR was defined as the presence of one or more surveillance drug resistance mutation according to the WHO surveillance list. Genotypic resistance to ARV was evaluated with Stanford HIVdb v7.0. Patterns of TDR, ADR and the prevalence of mutations over time were analyzed using logistic regression. Results and Discussion: The prevalence of TDR increased from 7.9% in 2003 to 13.1% in 2017 (*p* < 0.001). This was due to a significant increase in both resistance to nucleotide reverse transcriptase inhibitors (NRTIs) and non-nucleotide reverse transcriptase inhibitors (NNRTIs), from 5.6% to 6.7% (*p* = 0.002) and 2.9% to 8.9% (*p* < 0.001), respectively. TDR was associated with infection with subtype B, and with lower viral load levels (*p* < 0.05). The prevalence of ADR declined from 86.6% in 2001 to 51.0% in 2017 (*p* < 0.001), caused by decreasing drug resistance to all antiretroviral (ARV) classes (*p* < 0.001). Conclusions: While ADR has been decreasing since 2001, TDR has been increasing, reaching a value of 13.1% by the end of 2017. It is urgently necessary to develop public health programs to monitor the levels and patterns of TDR in newly diagnosed patients.

## 1. Introduction

In 2014, the WHO proposed the 90–90–90, an ambitious target to help end the AIDS pandemic in 2020—that 90% of people living with HIV are diagnosed, that 90% of those diagnosed are on treatment, and that 90% of the people on treatment are virally suppressed [1]. In Portugal, a national report estimated that in 2017, 39,820 persons were living with HIV. However, Portugal achieved the 90–90–90 goals in 2017—92.2% of people living with HIV were already diagnosed, 90.3% of those were on antiretroviral treatment and 93.0% of those were virally suppressed [2].

A major obstacle to achieving the 90–90–90 target is ARV drug resistance. While antiretroviral therapy (ART) largely decreases HIV-related morbidity and mortality, improves the quality of life of HIV-1 infected patients and reduces the risk of onward transmission [3,4,5], its scale up can potentiate the risk of the development of ARV drug resistance.

International guidelines consistently recommend that newly diagnosed individuals should be tested for ARV drug resistance, to detect potential transmitted drug resistance (TDR) and guide the selection of ART regimens [6,7]. This procedure minimizes the risk of experiencing virologic failure after starting ART due to the selection of resistant strains. In Portugal, the first line regimen recommends the use of an integrase strand inhibitor (INSTI), such as dolutegravir (DTG), raltegravir (RAL) or evitelgravir (EVG/c), together with a combination of nucleoside reverse transcriptase inhibitors (NRTI), such as tenofovir/entricitabine (TDF/FTC) or abacavir/lamivudine (ABC/3TC). Another option is to use a non-nucleoside reverse transcriptase inhibitor (NNRTI), such as rilpivirine (RPV), together with a combination of NRTI (TDF/FTC or ABC/3TC).

The active surveillance of TDR and ADR is crucial to understand factors involved in the transmission of HIV-1 drug resistance, and also to help to design effective ART treatment guidelines in different epidemic settings. Drug resistance evolves dynamically, and therefore it is extremely important to monitor temporal trends in TDR and ADR.

In this study, we aim to describe the temporal trends of TDR and ADR between 2001 and 2017, as well as the most prevalent drug resistance mutations, and to identify predictors of TDR among HIV-1 infected patients treated in Portuguese hospitals.

## 2. Material and Methods

### 2.1. Study Population

The protocol was in accordance with the declaration of Helsinki and approved by the Ethical Committee of Centro Hospitalar de Lisboa Ocidental (CHLO) (108/CES-2014; October 13rd 2014). The Portuguese HIV-1 resistance database contains anonymized patients’ information, including demographic, clinical and genotype resistance data from patients followed up in 22 hospitals located countrywide and collected between May 2001 and December 2017. All patients’ data were generated in the context of routine clinical care and collected in RegaDB [8]. RegaDB is a free and open source data management and analysis environment for infectious diseases, which allows clinicians to store, manage and analyze patient data, including viral genetic sequences. The study included individuals aged 18 years old or above and who had had an HIV-1 drug resistance test performed. Patients’ viral sequences were categorized into ART-naïve patients (ART-NP) and ART-experienced patients (ART-EP), by comparing the date of the first drug resistance test and the date of the first antiretroviral prescription. The CD4 cell count and HIV-1 viral load measured closest to the date of the drug resistance test were included in the analysis.

### 2.2. Drug Resistance Analyses and Subtyping

The genomic data included protease and reverse transcriptase sequences obtained through population sequencing completed at Laboratório de Biologia Molecular of Hospital de Egas Moniz/CHLO. Viral sequences were considered to originate from ART-NP when the first ART regimen was started simultaneously with or after the date of sample collection for drug resistance testing. Only the first HIV genotypic resistance test per patient was considered for the estimation of TDR for ART-NP. Furthermore, only the first resistance test after the first suspected virological failure was used for the estimation of ADR in ART-EP.

TDR was defined as the presence of one or more surveillance drug resistance mutation (SDRM) according to the WHO 2009 surveillance list [9]. Nucleotide sequences were submitted to the Calibrated Population Resistance tool version 8.0. Clinical resistance to ARV drugs was inferred using the Stanford HIVdb v8.4. Using this algorithm, ADR mutations were also identified. HIV-1 subtypes and circulating recombinant forms (CRF) were determined as previously described [10,11].

### 2.3. Statistical Analysis

Proportions and confidence intervals for proportions were calculated using 95% Wilson confidence interval for binomially distributed data. The differences between the prevalence of resistance in naïve and treated patients were analyzed using the Mann–Whitney U test and the X^2^ tests. Logistic regression was used to examine the association between demographic and clinical factors and the occurrence of SDRMs, and to analyze trends over time. For statistical analysis we considered a 5% significance level. All analyses were conducted in SPSS Statistic version 25 software and R3.5.1. More information in Appendix A.

## 3. Results

### 3.1. Population

Of 12,792 patients included in the database, 11,912 (93.1%) presented viral sequences. More than half (7310, 61.4%) were ART-NP, and 3848 (32.3%) were ART-EP. ART status was not available for 754 (6.3%) patients.

### 3.2. Characteristics of Portuguese Population

The characteristics of the study population are presented in Table 1. The ART-NPs were predominantly male (64.6%). The median age at the time point of HIV genotyping resistance testing was 38.0 years. The main mode of transmission was heterosexual contact (4.9%), followed by homosexual contact (2.6%) and intravenous drug use (2.6%). Of the ART-EP, 67.3% were male, presenting a median age of 39.0 years at the time of the drug resistance test. Heterosexual contact (4.7%) was the most common mode of transmission, followed by intravenous drug use (4.5%) and homosexual contact (1.8%). Both groups of patients, ART-NP and ART-EP, were predominantly infected with subtype B (36.7% and 42.6%), followed by subtype G (25.2% and 32.8%) and Circulating Recombinants Forms (CRFs) (22.2% and 16.5%). Most patients were born in Portugal (34.7% and 29.9% for ART-NP and ART-EP, respectively). Patients born abroad mostly originated from Sub-Saharan Africa (13.3% and 11.1% for ART-NP and ART-EP, respectively). At the time of HIV genotypic resistance testing, the median viral load (VL) for ART-NP was 4.8 Log_10_ copies/mL, and the CD4 cell count was 321.5 cells/mL. The median viral load for treated patients was 4.2 Log_10_ copies/mL and the CD4 cell count was 262 cells/mL .

### 3.3. Transmitted HIV Drug Resistance (TDR)

Overall, the prevalence of TDR between 2001 and 2017 was 9.4% (95%CI: 8.8–10.1%). Nucleoside reverse transcriptase inhibitor (NRTI) mutations were detected in 4.0% (95%CI: 3.5–4.4%) of ART-NP, non-nucleoside reverse transcriptase inhibitor (NNRTI) mutations in 5.0% (95%CI: 4.5–5.5%), and protease inhibitors (PI) in 2.8% (95%CI: 2.5–3.2%). In total, 7.3% (95%CI: 6.7–7.9%) presented single class resistance, 1.9% (95%CI: 1.6–2.2%) dual class resistance and 0.2% (95%CI: 0.1–0.4%) triple class resistance (Table 2).

Trends for TDR were determined only between the years 2003 and 2017, since the number of patients tested for ARV drug resistance between 2001 and 2002 was not large enough to allow for robustness in the statistical analyses.

TDR presented a significantly increasing trend from 7.9% in 2003 to 13.1% in 2017 (*p*_for-trend_ < 0.001). This trend was steeper in the last four years analyzed (2014 to 2017) (*p*_for-trend_ = 0.008). As the Portuguese recommendations for Treatment for All started in 2015 and the international Antiviral Society-USA Panel have recommended it since 2014, we used the time period between 2014 and 2017 [12] to analyze the potential effect of Treatment for All on drug resistance levels. This increasing trend was significant for two drug classes: NRTIs (5.6% in 2003 to 6.7% in 2017, *p*_for-trend_ = 0.002) and NNRTIs (2.9% in 2003 to 8.9% in 2017, *p*_for-trend_ < 0.001). For PIs, on the other hand, there was a trend of lower TDR levels in more recent years (4.0% in 2003 to 2.2% in 2017). However, this was not significant (*p*_for-trend_ = 0.985) (Figure 1A, Table 2).

Single-class mutations increased over time, from 4.4% (2003) to 10.0% (2017), as did double-class (2.8% to 3.7%) (*p*_for-trend_ = 0.002 for single-class and *p*_for-trend_ = 0.005 for double-class). Double-class resistance to NRTIs/NNRTIs combinations presented a significance increase (0.08% in 2003 to 3.2% in 2017; *p*_for-trend_ < 0.001), whereas NRTIs/PIs combinations presented a significant decrease (2.0% in 2003 to 0.3% in 2017; *p*_for-trend_ = 0.021) and NNRTIs/PIs combination presented a boarderline decrease (0.0% in 2003 to 0.2% in 2017, *p*_for-trend_ = 0.994). For the time period between 2014 and 2017, TDR, NRTIs and NNRTIs presented an increase (*p*_for-trend_ = 0.008 for TDR, *p*_for-trend_ = 0.001 for NRTIs and *p*_for-trend_ = 0.044 for NNRTIs), with a higher growth rate when compared to the previous period of time. On the other hand, NRTIs/PIs and NNRTIs/PIs combinations switched the trend, from a decrease to an increase, although this trend was not significant (*p*_for-trend_ = 0.950 for NRTIs/PIs and *p*_for-trend_ = 0.350 for NNRTIs/PIs) (Table 2 and Appendix A).

Transmitted drug resistance was also calculated considering the NNRTI mutation E138A/K, which confers resistance to rilpivirine. While this mutation is not yet included in the 2009 WHO list, it is extremely important to analyze it in TDR, given the recommended use of rilpivirine as a first line treatment. When considering E138K mutation, TDR increased from 9.4% (95%CI: 8.8–10.1) to 9.5% (95%CI: 8.9–1.0%), while NNRTIs TDR increased from 5.0% (95%CI: 4.5–5.5%) to 5.1% (95%CI: 4.6–5.7%). However, when the E138A polymorphism was analyzed separately, TDR increased to 11.9% (95%CI: 11.3–12.8) and NNRTIs TDR increased to 7.6% (95%CI: 7.0–8.2%) According to the Standford database, E138A is a polymorphic mutation that ranges in prevalence between 1% and 5%, depending on the background subtype, and it reduces RPV susceptibility by about twofold. Since the current Portuguese guidelines recommend the use of rilpivirine as the preferential NNRTI combined with two NRTI as first line regimen, when the patient presents a viral load <100,000 copies/mL, we think that is important to do this analysis in the Portuguese context.

The most frequently detected mutations were K103NS (3.2%) conferring resistance to NNRTIs, followed by M41L (1.6%) and M184V/I mutations (1.3%) conferring resistance to NRTIs, and L90M (1.2%) conferring resistance to PIs (Figure 2A).

### 3.4. Acquired HIV Drug Resistance (ADR)

The prevalence of ADR was 69.0% (95%CI: 67.6–70.5%). NRTIs resistance mutations were predominantly identified (57.8%, 95%CI: 56.2–59.4%), followed by NNRTIs mutations (45.8%, 95%CI: 44.2–47.4%) and PIs (23.6%, 95%CI: 22.3–25.0%). In total, 20.7% of patients (95%CI: 19.0–22.0%) had single-class resistance, 38.0% (95%CI: 37.0–40.0%) had dual-class resistance and 9.9% (95%CI: 9.0–10.9%) had triple-class resistance (Figure 1B and Table 2).

Overall, ADR decreased over time, from 86.6 to 50.9 between 2001 and 2017 (*p*_for-trend_ < 0.001). This decreasing trend was maintained, although was not significant after the treatments for all recommendations were implemented (2014 to 2017; *p*_for-trend_ = 0.837). Resistance to all drug classes decreased over time (2001–2017), as follows: resistance to NRTIs dropped from 80.8% to 33.3% (*p*_for-trend_ < 0.001); to NNRTIs from 46.4% to 40.2% (*p*_for-trend_ < 0.001) and to PIs from 52.7% to 10.8% (*p*_for trend_ < 0.001). For all these classes, for the 2014 to 2017 time period, the decreasing trend was maintained, however without a significant p for trend (*p*_for-trend_ = 0.874 for NRTIs, *p*_for-trend_ = 0.852 for NNRTIs and *p*_for-trend_ = 0.669 for PIs) (Figure 1Band Table 2, and Appendix A).

For double- and triple-class ADR resistance, significantly decreasing trends were also observed (*p*_for-trend_ < 0.001 for double class; *p*_for-trend_ < 0.001 for triple class), whereas for single class there was an increasing trend (*p*_for-trend_ = 0.019). For combinations of antiretroviral classes—NRTIs/NNRTIs (*p*_for-trend_ = 0.002), NRTIs/PIs (*p*_for-trend_ < 0.001) and NNRTIs/PIs (*p*_for-trend_ = 0.405)—decreasing trends were observed between 2001 and 2017 (Table 2 and Appendix A).

The most prevalent mutations conferring resistance to NRTIs in treated patients were M184IV (45.3%) and TAMs, such as T215YF (17.4%) and M41L (16.0%). K103NS (26.0%) and L90M (11.3%) were the most prevalent mutations conferring resistance to NNRTIs and to PIs, respectively (Figure 2B)

Trends for the frequencies of specific drug resistance mutations were also analyzed between 2001 and 2017, when mutations had a prevalence greater than 0.1% for ART-naive patients and 1.0% for ART-EP. Among ART-NP, L90M and D67NGE mutations presented a significant decline over the years (*p*_for-trend_ = 0.002 and *p*_for-trend_ = 0.005, respectively), and K103NS presented an increasing rate over the years (*p*_for-trend_ = 0.041). Although the trend for M184V was not significant (*p*_for-trend_ = 0.473), we observed a consistent increasing trend after 2012 (Figure 2C). For ART-experienced patients, on the other hand, PI resistance mutations M46IL, I54VLMTAS, V82ATSF and L90M significantly decreased over time (*p*_for-trend_ < 0.001). The same was observed for all mutations analyzed for NRTIs regimens (*p*_for-trend_ < 0.001), except for M184IV (*p*_for-trend_ = 0.639) and L74IV (*p*_for-trend_ = 0.379). For NNRTIs, only K103NS presented a significant trend (*p*_for-trend_ = 0.020) (Figure 2D and Appendix A).

### 3.5. Drug Susceptibility

According to the HIVdb Stanford database algorithm, 6.6% of ART-NP (7310) presented high-level resistance to at least one drug. In total, 5.4% of those had high-level resistance to a drug recommended for first line treatment. NNRTIs presented the highest level of high-level resistance (4.9%), with nevirapine (NVP) having the highest proportion of high-level resistance (4.7%), followed by efavirenz (EFV) with 4.0%. High-level resistance to NRTIs occurred in 1.7% of patients, with emtricitabine (FTC) and lamivudine (3TC) presenting the highest level (1.3%). High-level resistance to PIs was found in 1.8% of patients, with atazanavir (ATV) presenting 0.3% of high-level resistance (Figure 3A).

Out of 3848 ART-EP, 64.0% had high-level resistance to at least one ARV drug class. However, considering only the drugs actually used as first line therapy, high-level resistance decreased to 62.1%. For these patients, high-level resistance to NRTIs was the highest (54.4%), with the highest values of high-level resistance to FTC and 3TC (45.3%). In total, 40.5% of patients presented high-level resistance to NNRTIs, with 40.1% presenting high-level resistance to NVP and 35% to EFV. The high-level resistance to PIs was 19.9%, with ATV presenting the highest value (9.6%) (Figure 3B).

Integrase strand transfer inhibitors (INSTIs) drug resistance was also analyzed, but only for patients treated with INSTIs, since integrase drug resistance testing in ART-NP patients is not yet recommended by the Portuguese health authorities. In patients previously treated with INSTIs that were tested for DR, we observed 37.3% of INSTI resistance. In total, 2.7% of patients presented high-level resistance to dolutegravir (DTG) and bictegravir (BIC), 28.0% presented high-level resistance to elvitegravir (EVG) and 32.0% high-level resistance to raltegravir (RAL). We also observed 8.0% of intermediate level resistance and 24.0% of low-level resistance to DTG. The most frequently observed mutation was N155H, which occurred in 57.1% of patients that presented INSTIs resistance.

### 3.6. Predictors of TDR

The clinical and socio-demographic factors significantly associated with TDR in the univariate model were infection with subtype B (as compared to non-B subtypes grouped together) and the logVL, while sex presented borderline significance—although not reaching conventional statistical significance (*p* < 0.05), male sex was associated with higher levels of TDR. Infection with subtype B and logVL above 5.1 were significantly associated with NRTIs drug resistance. Male sex, origin from the Sub-Saharan Africa region, infection with subtype B and logVL above 5.1 were significantly associated with PIs drug resistance, and logVL above 5.1 was significantly associated with NNRTIs drug resistance (Appendix A).

Multivariate analysis indicated that TDR and NRTIs drug resistance were significantly associated with infection with subtype B (compared to non-Bs grouped together) and with logVL higher than 5.1 (Table 3).

We further observed that ART-NP with logVL above 5.1 presented significantly lower levels of K103NS and L90M mutations (*p* = 0.023 and *p* = 0.008, respectively), while patients with a logVL above 4.1 presented lower levels of M184V (*p* < 0.001) (Appendix A).

### 3.7. Predictors of ADR

The univariate logistic regression analysis exploring predictors of acquired drug resistance is shown in Appendix A. Male sex, age at diagnosis above 26 years old, MSM transmission group (compared to being heterosexual), origin from Europe, infection with subtype B, and a logVL above 4.1 were all significantly associated with ADR (Appendix A).

Multivariate analysis showed that any ADR and NRTIs drug resistances were significantly associated with male sex, infection with subtype B and a logVL above 4.1 (Table 3).

## 4. Discussion

Our study showed that the estimated prevalence of TDR over time, among ART-NP, increased between 2003 and 2017, and this trend was more pronounced between 2014 and 2017. While TDR to NRTIs and NNRTIs increased, TDR to PIs decreased. We hypothesize that this increase could have different explanations. First, given that the increasing trend for TDR is steeper after 2014, this could be correlated with the implementation of the Treatment for All recommendations. Second is the changing face of the pandemic, with an increasing proportion of MSMs among new diagnoses, which are associated with the faster onward transmission of HIV infection with less reversion of DRM and therefore the potentiation of transmission of TDR. Moreover, the MSM group are the most frequently tested, having an earlier diagnosis, which may imply less reversion of mutations, increasing TDR detection. Finally is the increasing mobility of populations, with the presence in Portugal of a high number of migrants from Portuguese-speaking Sub-Saharan African countries where TDR has been increasing in the last years [13,14,15,16].

Similarly to our study, a study of HIV drug resistance in a Canadian cohort showed an increase in TDR between 1996 and 2014, with a more prominent increase in the last years [16], which is consistent with our results. Some other studies, however, showed the opposite results, with TDR prevalence decreasing or stabilizing over time [17,18,19,20,21]. A potential explanation for our discrepant results could be the origin of migrants living in Portugal who come from Portuguese-speaking Sub-Saharan African countries where TDR has been increasing in the last years [13,14,15,16].

When looking at specific drug classes, we find an increasing trend of TDR to NRTIs and NNRTIs, but a decreasing trend of TDR to PIs. We hypothesize that this could be due to patterns of drug usage, with an infrequent use of PIs as first line treatment in Portugal. Another explanation could be the larger flexibility and tolerability of protease to mutations, which could imply a lower probability of fixation of mutations.

The most frequently detected mutations in ART-NP were K103NS, which confers high-level resistance to NVP and EFV, M41L, which reduces susceptibility to TDF and ABC when in combination with other NRTIs mutations, and M184VI, which causes high-level resistance to recommended first line drugs 3TC and FTC [22]. L90M was the resistance mutation to PIs with the highest prevalence, and it causes reduced susceptibility to ATV and LPV. M41L and M184IV have an impact on susceptibility to drugs recommended for first line use, so it is important to understand the impact of the transmission of these mutations and why its prevalence is increasing among ART-NP (Figure 3). E138A and E138K were also considered separately, since they confer resistance to rilpivirine, which is included in the preferential regimens used in Portugal. These polymorphisms/mutations were present in ART-NP, which is consistent with other studies [23,24]. By not considering this codon, given that the 2009 WHO SDRMs list is still largely used, the TDR levels reported in other studies may be underestimated.

The risk of TDR was significantly lower in patients infected with non-B subtypes compared to subtype B, as previously shown [25,26,27,28]. We hypothesize that this could be due to the longer use of ARV therapy in developed countries, where this subtype is more prevalent, and to the circulation of this subtype in MSMs, where transmission chains are faster and therefore potentiate the transmission of TDR. However, there are features of the changing HIV-1 epidemic that could lead to an inversion of this pattern. The Treatment for All, the increasing prevalence of non-B subtypes in developed countries, and, particularly, the growing number of reports indicating transmission clusters of non-B subtypes in MSM, could lead to an increase in TDR in non-B subtypes.

Another ongoing study at our lab, comparing a cohort of late presenters with non-late presenters, interestingly showed that individuals involved in transmission clusters were more frequently non-late presenters, and more frequently presented TDR [29]. This suggests that mutations associated with resistance are maintained and transmitted forward in these clusters, indicating that they do not reduce viral transmissibility. Previous studies have reported that TDR is mostly due to onward transmission between ART-NP [30,31,32], and another study has also shown the persistence and high effective reproductive number (Re) of specific NNRTI resistance mutations (K103N and E138A) in local transmission networks in Greece [33]. Yet another study, however, indicated that, among several studied DRMs (41L, 67N, 70R, 184V, 210W, 215D, 215S and 219Q (nRTI-related), and 103N, 108I, 138A, 181C and 190A), only the L90M mutation in the protease gene was found to have significantly higher fitness than the drug-sensitive strains [34].

Interestingly and paradoxically in relation with the previous findings of the maintained transmissibility of these TDR strains, ART-NP diagnosed with M184V, L90M and K103N TDR mutations presented statistically significant lower levels of VL than ART-NP without DRMs. This is not surprising for M184V, given the previously reported high fitness cost of this mutation [13,20,34,35,36]. For K103NS, no statistical association had been found previously, which could be due to its lower impact on fitness compared to M184V [37,38]. While we would expect that this lower VL should indicate a low transmissibility, our finding of an increasing trend in the prevalences of M184V and K103NS in the last few years, and a higher prevalence of TDR inside transmission clusters, indicate that the relationship between VL and transmissibility should be further elucidated. Indeed, the transmissibility of HIV strains should be determined not only by the VL, but by a trade-off with other aspects, such as infectiousness and disease progression. Other studies have shown that specific TDR mutations have an effect on viral fitness that can have implications for its transmissibility [39,40,41]. Our consistent finding of lower VLs in patients carrying M184V and K103N, despite increasing prevalence, warrants future directed investigations of these results.

Strikingly, in the same time period (2001–2017) we found a significant decreasing trend in ADR that spans all drug classes. This trend has been consistently found in other studies. For example, in a multi-center cohort (1997–2008) in Switzerland, it was demonstrated that the majority of treated patients who initiated treatment in more recent years did not acquire drug resistance [36], as well as another study in a large Canadian cohort (1996–2016), which showed that the prevalence of ADR has been decreasing for all drug categories [16]. Concordant results were also observed in an Italian cohort (2003–2012), in a German cohort (2001–2011), in Spain (1999–2005) and in western Europe (1997–2008) [17,35,42,43]. Rocheleau et al. (2017) proposed that this decreasing trend of ADR is due to a combination of factors, which include the increased efficacy of ARV regimens, readily accessible combination regimens, and improved patient management [16].

For ART-experienced patients, the more prevalent mutations conferring resistance to NRTIs were M184IV, conferring high-level resistance to FTC and 3TC, and also reduced susceptibility to ABC. K103NS was the most prevalent mutation conferring resistance to NNRTIs, specifically to NVP and EFV, and L90M reduced susceptibility specifically to ATV and LPV. Over time we observed an important reduction in specific PIs and NRTIs mutations, such as M46IL, L90M, M41L and L210W. This could be associated with the higher genetic barrier of boosted PIs and of dual NRTIs formulations (2014) [44]. Moreover, antiretroviral regimens changed over time, with a large reduction in the prescription of thymidine analogues in Europe, which surely associates with the observed decrease in TAMs.

Similarly to TDR, ADR patients tended to present lower levels of VL at the time of resistance testing, which is consistent with the findings of the Swiss cohort [45]. We also found that the risk of ADR was significantly higher in male patients [17] and in patients infected with subtype B. ADR can be related to poor adherence, which has been shown to be a major determinant of virologic failure and the emergence of drug-resistant viruses. Barriers to optimal adherence may originate from individual (biological, socio-cultural, behavioral), pharmacological and societal factors, which makes these findings more complex to interpret. Other studies also associated the development of ADR with levels of adherence in each era of therapy initiation [16,17]. On the other hand, factors found to be associated with ADR—given its decreasing prevalence, and therefore the lower number of patients diagnosed with ADR in more recent years—could reflect older features of ADR acquisition, and not the most recent. Finally, subtype B, which caused the initial pandemic in the Western World, has been subject to selective pressure by ARV drugs for much longer than the other subtypes, and this should explain why we find significantly more ADR in this subtype.

## 5. Conclusions

In summary, our study showed that while ADR has been steadily decreasing since 2001, TDR has been increasing, reaching a worrying value of 13.1% in 2017. While the decreasing ADR seems to be caused mainly by the increasing efficacy of ARV therapy, TDR seems to be mainly driven by determinants of the virus, such as viral subtype and the fitness of the virus in the presence of particular mutations. Our results highlight that it is urgent to develop public health programs to monitor the levels and patterns of TDR in newly diagnosed patients.

## Figures and Tables

**Figure 1 viruses-12-01238-f001:**
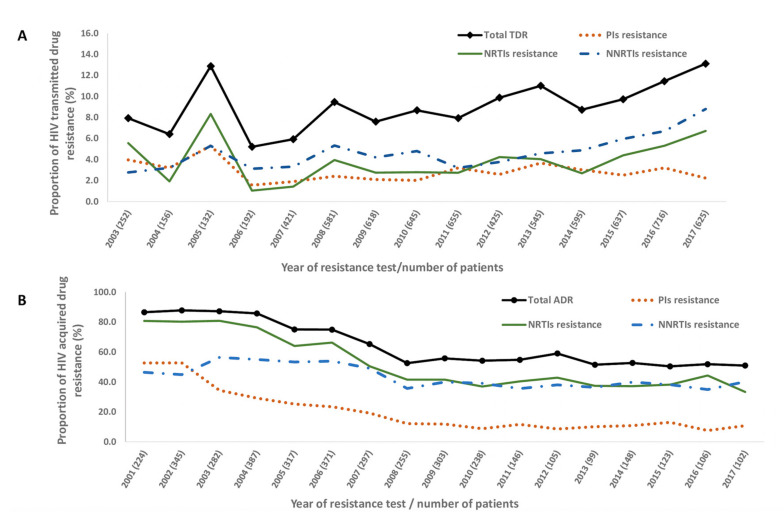
Proportion of (**A**) transmitted drug resistance (TDR) in sequences from ART-NP between 2003 and 2017 and (**B**) of acquired drug resistance (ADR) in ART-EP between 2001 and 2017. NRTI, nucleotide reverse transcriptase inhibitors; NNRTI, non-nucleotide reverse transcriptase inhibitors; PI, protease inhibitors; ART-NP, antiretroviral-naïve patients; ART-EP, antiretroviral-experienced patients.

**Figure 2 viruses-12-01238-f002:**
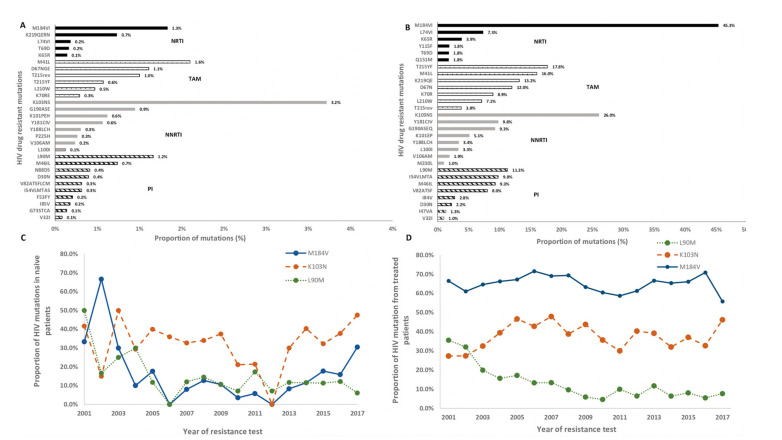
Proportion of resistance mutations in sequences (**A**) ART-NP and (**B**) ART-EP and proportion of M184V, K103N and L90M mutations in (**C**) ART-NP and (**D**) ART-EP over time between 2001 and 2017. TAM, Thymidine analog mutation; NRTI, nucleotide reverse transcriptase inhibitors; NNRTI, non-nucleotide reverse transcriptase inhibitors; PI, protease inhibitors. ART-NP, antiretroviral-naïve patients; ART-EP, antiretroviral-experienced patients.

**Figure 3 viruses-12-01238-f003:**
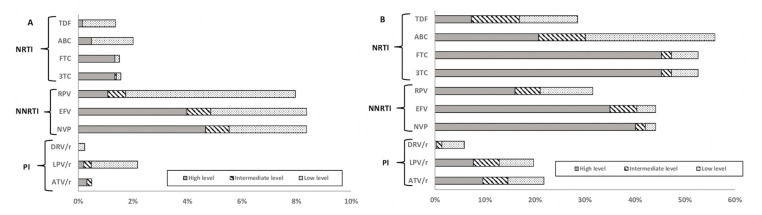
Predicted phenotypic resistance (Stanford scores) for antiretroviral drugs currently recommended as first line therapy in Portugal (**A**) for ART-NP (2003–2017) and (**B**) for ART-EP (2003–2017). Scores of low-level (score 2 and 3), intermediate level (score 4) or high-level (score 5) resistance were used to predict phenotypic resistance. Abbreviations: NRTI, nucleoside reverse transcriptase inhibitors; NNRTI, non-nucleoside reverse transcriptase inhibitors; PI, protease inhibitors; FTC, emtricitabine; TDF, tenofovir; 3TC, lamivudine; ABC, abacavir; EFV, efavirenz; RPV, rilpivirine; DRV/r, darunavir; LPV/r, lopinavir; ATV/r, atazanavir; ART-NP, antiretroviral-naïve patients; ART-EP, antiretroviral-experienced patients.

**Table 1 viruses-12-01238-t001:** Demographic and clinic patient characteristics, 2001–2017. ART, antiretroviral; ART-NP, antiretroviral-naïve patients; ART-EP, antiretroviral-experienced patients.

Patient Characteristics	ART-NP	ART-EP
**Total, n (%)**	7310 (100%)	3848 (100%)
**Sex, n (%)**		
Female	2521 (34.5%)	1239 (32.2%)
Male	4719 (64.6%)	2588 (67.3%)
**Median age at genotyping in years (IQR)**	38.0 (31.0–48.0)	39.0 (33.0–46.0)
18–25	766 (10.5%)	153 (4.0%)
26–40	3287 (45.0%)	1930 (50.2%)
41–55	2263 (31.0%)	1366 (35.5%)
>56	843 (11.5%)	340 (8.8%)
**Mode of transmission, n (%)**		
Heterosexuals	358 (4.9%)	181 (4.7%)
Men who have sex with men	193 (2.6%)	69 (1.8%)
Intravenous drug use	189 (2.6%)	173 (4.5%)
Others	78 (1.1%)	106 (2.8%)
Unknown	6492 (88.8%)	3319 (86.3%)
**Region of origin, n (%)**		
Portugal	2535 (34.7%)	1152 (29.9%)
Sub-Saharan Africa	972 (13.3%)	427 (11.1%)
South America	201 (2.8%)	46 (1.2%)
Europe	81 (1.1%)	20 (0.5%)
Others	10 (0.1%)	4 (0.1%)
Unknown	3511 (48.0%)	2199 (57.2%)
**Year of genotyping, n (%)**		
2001–2005	655 (9.0%)	1555 (40.4%)
2006–2009	1812 (24.8%)	1226 (31.9%)
2010–2013	2270 (31.1%)	588 (15.3%)
2014–2017	2573 (35.2%)	479 (12.4%)
**HIV-1 subtype, n (%)**		
Subtype B	2686 (36.7%)	1639 (42.6%)
Subtype G	1845 (25.2%)	1262 (32.8%)
Subtype C	499 (6.8%)	149 (3.9%)
Subtype A	312 (4.3%)	55 (1.4%)
Subtype F1	265 (3.6%)	72 (1.9%)
Circulating Recombinants Forms (CRFs)	1621 (22.2%)	635 (16.5%)
Other HIV-1 subtypes	67 (0.92%)	28 (0.7%)
**CD4 count at time of resistance test** (cells/mL)		
Median CD4 count (IQR; range)	321.5 (145.0–505.0) (0.0–1905.0)	262.0 (136.0–440.0) (0.0–1844.0)
<50	415 (5.7%)	308 (8.0%)
51–200	762 (10.4%)	868 (22.6%)
201–350	807 (11.0%)	817 (21.2%)
351–500	735 (10.1%)	533 (13.9%)
501	929 (12.7%)	565 (14.7%)
Unknown	3662 (50.1%)	757 (19.7%)
**Viral Load at time of resistance test** (log_10_ copies/mL)		
Median Log Viral Load (IQR)	4.8 (4.2–5.4)	4.2 (3.5–4.8)
<4.0	1111 (15.2%)	1526 (39.7%)
4.1 to 5.0	2354 (32.2%)	1227 (31.9%)
>5.1	2187 (29.9%)	620 (16.1%)
Unknown	1658 (22.7%)	475 (12.3%)

**Table 2 viruses-12-01238-t002:** Proportion of transmitted drug (TDR) and of acquired drug resistance (ADR) between 2001 and 2017. ART-NP, antiretroviral-naïve patients; ART-EP, antiretroviral-experienced patients; P-value for trend of TDR between 2003 and 2017 and of ADR between 2001 and 2017. Prot, protease; RT, reverse transcriptase; DRM, drug resistance mutations; NRTI, nucleotide reverse transcriptase inhibitors; NNRTI, non-nucleotide reverse transcriptase inhibitors; PI, protease inhibitors; CI, confidence interval; OR, odds ratio.

Transmitted Drug Resistance (TDR)	n (%)	95% CI	OR (95% CI)	*p* for Trend (2003–2017)
Prot/RT Sequence from ART-NP				
Total	7310 (100.0%)			
Any DRMs	690 (9.4%)		1.046 (1.024–1.068)	<0.001
NRTI resistance	289 (4.0%)	3.5–4.4	1.053 (1.019–1.088)	0.002
NNRTI resistance	367 (5.0%)	4.5–5.5	1.053 (1.028–1.078)	<0.001
PI resistance	206 (2.8%)	2.5–3.2	1.000 (0.964–1.038)	0.985
Single class resistance	535 (7.3%)	6.7–7.9	1.038 (1.014–1.063)	0.002
Dual class resistance	138 (1.9%)	1.6–2.2	1.071 (1.021–1.123)	0.005
PI + NRTI resistance	30 (0.4%)	0.3–0.6	0.891 (0.808–0.982)	0.021
PI + NNRTI resistance	18 (0.2%)	0.2–0.4	0.999 (0.883–1.132)	0.994
NRTI+NNRTI resistance	90 (1.2%)	0.9-1.5	1.163 (1.091–1.240)	<0.001
Triple class resistance	17 (0.2%)	0.1–0.4	1.019 (0.895–1.160)	0.779
**Acquired drug resistance (ADR)**	**n (%)**	**95% CI**	**OR (95% CI)**	***p* for trend (2001–2017)**
**Prot/RT sequence from ART-EP**				
Total	3848 (100.0%)			
Any DRMs	2657 (69.0%)	67.6–70.5	0.867 (0.852–0.881)	<0.001
NRTI resistance	2225 (57.8%)	56.2–59.4	0.854 (0.840–0.868)	<0.001
NNRTI resistance	1763 (45.8%)	44.2–47.4	0.952 (0.938–0.967)	<0.001
PI resistance	909 (23.6%)	22.3–25.0	0.822 (0.804–0.841)	<0.001
Single class resistance	798 (20.7%)	19.0–22.0	1.022 (1.004–1.040)	0.019
Dual class resistance	1478 (38.0%)	37.0–40.0	0.908 (0.893–0.923)	<0.001
PI + NRTI resistance	449 (11.6%)	10.6-12.7	0.813 (0.788–0.839)	<0.001
PI + NNRTI resistance	24 (0.6%)	0.4–0.9	0.959 (0.870–1.058)	0.405
NRTI+NNRTI resistance	1005 (26.0%)	24.7–27.5	0.974 (0.957–0.990)	0.002
Triple class resistance	381 (9.9%)	9.0–10.9	0.840 (0.814–0.867)	<0.001

**Table 3 viruses-12-01238-t003:** Multiple regression analysis of factors associated with HIV-transmitted drug resistance. TDR, transmitted drug resistance; ADR, acquired drug resistance; CI, confidence interval; OR, odds ratio. * *p* < 0.05.

	Any TDR		NRTI TDR		NNRTI TDR		PI TDR	
**Variable**	**OR (95%CI)**	***p***	**OR (95%CI)**	***p***	**OR (95%CI)**	***p***	**OR (95%CI)**	***p***
**Sex**								
Female *								
Male	1.21(0.95–1.55)	0.124	1.07(0.73–1.59)	0.708	1.03(0.75–1.40)	0.852	1.67(1.07–2.94)	0.024
**Age at diagnosis**								
18–25 *								
26–40	0.97(0.69–1.36)	0.860	1.33(0.72–2.44)	0.361	0.75(0.49–1.15)	0.186	1.78(0.91–3.50)	0.093
41–55	0.98(0.68–1.40)	0.896	1.74(0.63–3.07)	0.082	0.76(0.48–1.21)	0.248	1.44(0.70–2.94)	0.319
>56	1.15(0.74–1.79)	0.520	1.40(0.63–3.07)	0.405	1.20(0.71–2.03)	0.498	2.02(0.89–4.59)	0.091
**Subtypes**								
B *								
Non-B	0.74(0.60–0.92)	0.006	0.45(0.31–0.65)	<0.001	1.19(0.89–1.59)	0.240	0.65(0.45–0.94)	0.021
**Viral Load** (log_10_ copies/mL)								
< 4.0 *								
4.1 to 5.0	0.73(0.56–0.96)	0.024	0.42(0.28–0.63)	<0.001	0.70(0.49–1.00)	0.054	0.76(0.49–1.19)	0.234
>5.1	0.66(0.49–0.87)	0.004	0.38(0.25–0.59)	<0.001	0.65(0.44–0.94)	0.023	0.50(0.31–0.83)	0.007

	**Any ADR**		**NRTI ADR**		**NNRTI ADR**		**PI ADR**	
**Variable**	**OR (95%CI)**	***p***	**OR (95%CI)**	***p***	**OR (95%CI)**	***p***	**OR (95%CI)**	***p***
**Sex**								
Female *								
Male	1.44(1.20–1.73)	<0.001	1.40(1.18–1.67)	<0.001	1.15(0.97–1.35)	0.112	1.66(1.35–2.05)	<0.001
**Age at diagnosis**								
18–25 *								
26–40	1.12(0.73–1.73)	0.602	1.13(0.74–1.72)	0.565	0.85(0.57–1.27)	0.420	1.71(0.95–3.07)	0.072
41–55	1.07(0.69–1.67)	0.764	1.11(0.72–1.70)	0.626	0.83(0.55–1.25)	0.375	1.76(0.97–3.18)	0.062
>56	1.27(0.76–2.12)	0.367	1.37(0.84–2.24)	0.207	0.84(0.53–1.35)	0.481	2.65(1.40–5.02)	0.003
**Subtypes**								
B *								
Non-B	0.68(0.57–0.80)	<0.001	0.61(0.52–0.72)	<0.001	0.98(0.84–1.14)	0.789	0.64(0.54–0.77)	<0.001
**Viral Load** (log_10_ copies/mL)								
<4.0 *								
4.1 to 5.0	0.63(0.52–0.76)	<0.001	0.60(0.50–0.71)	<0.001	0.89(0.75–1.05)	0.165	0.85(0.70–1.03)	0.090
>5.1	0.34(0.27–0.42)	<0.001	0.29(0.23–0.36)	<0.001	0.69(0.56–0.85)	0.001	0.50(0.38–0.65)	<0.001

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
