# Peer review of "Increasing Prevalence of HIV-1 Transmitted Drug Resistance in Portugal: Implications for First Line Treatment Recommendations"

_viruses, 2020, doi:10.3390/v12111238_

Round 1

Reviewer 1 Report

This paper aims to describe changes over time in the genotypic resistance profiles of HIV-1-infected individuals in Portugal over a 17-year period from 2001. Patients are divided into treatment-naïve and -experienced, with trends in the former being largely restricted to the WHO surveillance mutations for Transmitted Drug Resistance (TDR) and the latter being a more complete examination of all resistance mutations – classed as Acquired Drug Resistance (ADR). Whilst this is a laudable exercise, and has precedent with similar studies of a number of other countries’ such trends (e.g. UK, Switzerland, Greece, etc.), there are serious flaws in the Methods, Results, and Discussion sections that need addressing before the manuscript can be accepted for publication.

As regards the methods, there are some general points that need raising about the nature of the study population, and changes over the time period that may account for the results but that have not been mentioned:

  • There is no information about treatment recommendations. Changes in resistance patterns (particularly ADR, but also TDR) will follow changes in treatment patterns, e.g.
    • the decline in NNRTI use has led to a drop in NNRTI resistance.
    • DRV is the first-line PI of choice in many countries, and LPV is not widely used in Europe any more. PI mutations reflect these changes
    • Similarly, the decline in TAMs is consistent with a dramatic decline in the use of thymidine analogues such as AZT, ddI, d4T etc.
  • It is not clear that the analysis only looked at the first resistance test per patient. Please clarify in the study population section of both the methods and the results. In the latter, patients are referred to, but it is the test that is being examined. This has implications when discussing time trends vis-à-vis demographic and subtype changes.
  • There is no information about the time between the resistance test, diagnosis and likely time of infection. The persistence or otherwise of resistance mutations and the VL at testing will be dependent upon these factors that are unlikely to be similar between cohorts.
  • Has there been a change in the patterns of resistance testing over time? Has it always been the case that new diagnoses are given a genotypic test as soon as possible? In previous years, it was often not performed at diagnosis, but pre-treatment, and this could be many months or years after the initial positive test. If the time between diagnosis and resistance test has shortened over the years, this may account for higher rates of TDR detection.
  • In a similar vein, has there been any change over the study period in the estimated time to diagnosis (i.e. estimated time between infection and diagnosis). If this has shortened (e.g. through more frequent testing, or targeting of high-risk cohorts), then the likelihood of TDR detection is also increased.
  • Line 85 – It is stated that “Phenotypic resistance…was evaluated”. This would imply in vitro testing of co-cultured or recombinant virus. What was done with Stanford is better expressed as “Clinical resistance…was inferred.”

The following points relate to the Results section (lines 96-252)

  • This paragraph starting at line 100 has too many verbose statistics and is effectively unreadable. It would be preferable for Table S1 to be in the main text instead.
  • Table 1 – The Odds Ratios are without meaning here as what they relate to is not mentioned. And the trend p-values are meaningless without stating the trend itself. Are the frequencies increasing or decreasing over time? By how much? What statistic is the p-value related to? The slope? Also, how can there be 48 patients with dual-class TDR, but 47, 35, and 107 with the three 2-member combinations of PI, NRTI and NNRTI TDR? That adds up to 189. Which seems very large. Even if the 17 triple-class TDRs were counted in each two-class statistic, this still doesn’t account for the data. The same problem occurs with ADR – 1,029 patients with dual-class resistance but 1,386 with NRTI & NNRTI resistance alone. This must simply be incorrect. The remaining analysis is fatally compromised by the data being unreliable.
  • Lines 131-139 – This trend analysis for TDR feels incorrect. Whilst there appears to be a trend for steadily increasing Total, NRTI and NNRTI TDR, I would be very surprised to learn that the trends were significant, let alone to the degree stated. For the area of the chart with the steepest upward slopes (2015-7) to then have an insignificant trend seems odd to say the least. The ADR data looks more trend-like, but still not very strong. Please could the authors explain in more detail how the trend p-statistics were derived, together with some sample data? This applies to a number of analyses further down the text.
  • Lines 147-150 – Why are the time intervals set at 2011-7 in the first analysis and 2014-7 in the second? These seem arbitrary when data since 2003 is available.
  • Lines 151-60 – The inclusion of E138A in TDR is misleading as is a common polymorphism in non-B subtypes and not indicative of transmitted resistance as a public health concern. Rather it only reinforces the utility of genotyping at diagnosis. It is far from certain that it would be included in a revised surveillance mutation list. 138K is not considered polymorphic, and re-analysing the trends when including both 138A & K is hence not useful as it does not accurately reflect TDR. Reference 31 has the same issue, and was explicit in its analysis. The relevance of E138K as a TDR would only be in the context of significant historical RPV use. Only if this is the case in Portugal would this section be merited. (Certainly in the ADR section it is of interest notwithstanding.)
  • Lines 171-5 – It makes no sense to talk of the 2003-17 cohort as a single entity here, given the emphasis on time trends. Similarly for the three classes.
  • Lines 176-187 – These trend analyses suffer from the same problems as the TDR ones. For example, the p-value for NNRTI detection is asserted to be <0.001 despite the change being minimal and the chart plot being essentially flat. The problems with the numbers in the table are already stated, and these data cannot be considered reliable. Trend analysis on the post-Treatment for All period is pointless given its brevity, and certainly cannot be meaningfully compared to the longer time trend. Multi-class analysis is similarly unsuitable.
  • Lines 192-8 – It seems that K103N and M184V are increasing in the drug-naïve cohort concomitant with a sharp decline in the experienced cohort. This needs exploring. As with other trend analyses, I am sceptical of the p-values associated with the individual mutations, particularly as there is no raw data to view.
  • Lines 200-213 – Much of the raw numbers are presented graphically (the first-line regimen data). Remove the detail around the total numbers and retain only headline figures.
  • Lines 224-9 – If true, the reported frequencies of acquired INSTI resistance are alarmingly and astonishingly large. Some information on the selection of these samples is essential, as it must be the case that these sequences derive solely from patients failing INSTI regimes.
  • Lines 231-43 – This area needs work:
    • Given the multivariate analysis, there seems to be no value in the univariate results. The significance of e.g. sex/subtype/origin/age at diagnosis etc. cannot be easily disentangled from each other.
    • Why are so few of the factors in Table S3 present in Table 2? They should all be included, if only to show non-significance.
    • The data on the individual mutations are not present in Table 2 but are referred to in the text.
    • Have the analyses been corrected for multiple comparisons (e.g. Holm-Bonferroni?
    • Replace “Gender” with “Sex” in Tables 2, S3 & S4.
  • Table S2 – How the OR and p­-values were obtained needs much more transparency. The choice of time intervals looks like p-mining. Why are there separate statistics for Sep 2014-2017 and for 2015-7?

Finally, there are issues with the Discussion. By focussing on demographic trends, the consequences of the likely temporal trends in testing and treatment mentioned above are overlooked.

  • Lines 257-8 – Please expand upon why TDR would be correlated with the Treatment for All recommendations? Aside from a temporal coincidence, what is the suggested mechanism by which the latter influenced the former? Increased testing and earlier detection? Refer to the Methods critique above to make sure all these putative causes are mentioned.
  • Lines 264-9 – Is the statement about increasing sub-Saharan migration consistent with the one about increasing MSM? Are both populations increasingly represented in the Portuguese HIV cohort, or is this an artefact of changing testing regimes over time? How can the sub-Saharan migrant population both have lower TDRs, as evidenced by the non-B rates in Table 2 (significantly lower for all but NNRTI), and simultaneously contribute to increased rates of TDR? How does the hypothesized impact of the rapid onward transmission suggested for the MSM cohort compare to the findings alluded to in lines 293-5 where TDR was found throughout a late-presenting cohort and associated with clusters of transmission? Why favour any of these hypotheses when the major question – “are the data a function of national treatment and testing protocols?” – is not raised? Does the inverse relationship between TDR and ADR in recent years point in one particular direction when looking at all these theories?
  • Lines 270-1 – E138A does not impact upon EFV or NVP, and K only weakly. The effects stated are for K103NS.
  • Line 273 – M184V does not cause high-level resistance to abacavir. Again, the syntax of the sentence is misleading.
  • Lines 274-5 & line 331 – L90M causes clinically significant PI resistance even in the absence of further mutations.
  • Lines 275-6 – Please remove "not that worrying" as it still precludes NNRTIs for the lifetime of the patient, even if it is not a first-line drug in 2020.
  • Lines 278-84 – I would downplay the E138AK results as they are of limited information given the background frequency of 138A in non-B subtypes. Unless the analysis ignored 138A and only explored 138K, their impact on the analyses should be minimised. As above, there is no suggestion that 138A would be in a revised SDRM list (although there is a stronger case for 138K).
  • Lines 285-92 – The impact of subtype on TDR prevalence as shown in Table 2 needs to be reconciled with this paragraph (see also lines 264-9).
  • Lines 299-303 – That E138A occurs as a natural polymorphism indicates it has minimal fitness impact, if any. Furthermore, how do the authors account for L being wild-type at position 90 of protease, if M confers “significantly higher fitness”? Either it doesn’t, and the referenced paper is mistaken, or there is a very odd phenomenon occurring here.
  • Lines 304-6 – I cannot see in the Results section the data supporting these claims about “significantly lower levels of VL” associated with the three mutations. Also, lower than what? Drug-naïve patients without these mutations? Individually or en masse?
  • Lines 304-317 – It is a major leap to go from an observed lower VL with certain mutations (that are in turn associated with other demographic and temporal characteristics) to discuss transmissibility, as if the latter was purely a function of VL at the moment of genotyping (irrespective of time since infection) and that the VL is substantially affected by these mutations directly. There is no data on the transmitting patients to support these claims. Phylogenetic methods can only partly mitigate this missing information. This paragraph needs to be looked at again, and probably downplayed considerably.
  • Line 320 – How can a multi-centre European cohort be in Switzerland?!
  • Line 318-327 – Given the dramatically improved suppression viral rates of ARV regimes over the last five years or so, together with better testing systems, just where is the TDR coming from? If it is asserted that it is arising from sub-Saharan migrants, then there needs to be more finely-resolved data presentation in the Results section around this.
  • Lines 328-334 – The hypotheses here need to be related to those made regarding several of these mutations and viral fitness (lines 293-317 roughly). Either way, the reduction in TAMs is almost certainly a consequence of prescribing patterns being radically different across the timeframe discussed. Thymidine analogues are very rarely prescribed in Europe these days, although they are still widely used in LMICs.
  • 335-345 – Firstly, can the relationship between male sex and subtype B virus truly be disentangled? Seeing as >85% of diagnoses have no transmission route, then essentially these are unlinked in the analyses. Secondly, does this not simply reflect time since infection? Most of these MSM subtype B viruses will have a long treatment history. Given the lack of clarity over the sampling, this remains unclear.

As a consequence of these major oversights, the Conclusion section needs a bit of revision.

The following points are minor points of composition:

  • Consider replacing DN / TP with treatment-naive / treatment-experienced, or ART-naive / ART-experienced. This has been done in Table S1 and should be used throughout
  • Express p-values as <10-x when it is below 0.001 - having a p-value of 0.000 is not appropriate. However, the trend analysis needs revisiting.
  • Lines 27, 133, 177, 290 – Capitalise or quote “Treatment for All” as per line 258.
  • Line 52 - expand ARV & ART at first use of the abbreviation.
  • Line 98 - remove the second instance of "patients"
  • Lines 103 or 107 - MSM used (correctly) on line 248 but not here. Please revise.
  • Lines 146, 155, 167, 332 - "over time" instead of "overtime"
  • Figure 2 – Several points:
    • The y-axes of 2A & 2B are incomplete – only half of the mutations are present.
    • The legends and axis titles need to be consistent (NRTIs vs. NRTI in 2A & 2B, and for 2C & 2D, patients vs. patient, and the order of mutations in the legends).
    • Remove the decimal place from the x-axis in 2B and from the y-axes of 2C & 2D.
  • Figure 3 - Remove the decimal place from the x-axes.
  • Lines 223-4 – There is a fullstop where the sentence should continue “…not yet recommended by health authorities.”
  • Line 227 – Replace “intermedium” with “intermediate”
  • Line 293 – This sentence needs tidying up.
  • Line 298 – “has also shown” for “have also shown”
  • Lines 300-3 – There is another rogue fullstop, here between ref 33 and only […190A) 33, only the…]
  • Line 332 – Too many “important”s.
  • Table S1 – Spelling for Sub-Saharan

Finally, the authors should not place the Tables and Figures inline with the text as it compromises the readability. It is better practice to collate them at the end of the manuscript.

Author Response

To the Board of editors.

We have recently submitted our manuscript entitled “Increasing prevalence of HIV-1 Transmitted Drug Resistance in Portugal: implications for first line treatment recommendations” to Viruses. We would like to acknowledge all the comments and suggestions provided by the journal editor and by the reviewers. We are now submitting the revised version, where we address all the comments and suggestions provided by the reviewers and by the journal editor. Overall, we feel that revision process has greatly contributed to a higher quality paper. A point-by-point answer is provided below. We hope to have answered the journal requirements and reviewers’ concerns and to provide a new version of the manuscript suitable for publication in Viruses.

Sincerely yours,

Marta Pingarilho

Reviewer 1

“There is no information about treatment recommendations. Changes in resistance patterns (particularly ADR, but also TDR) will follow changes in treatment patterns, e.g.”

We would like to thank the reviewer for the comment. In fact, this is something we have in mind in the course of our analysis. However, as there is no reliable information about changes in treatment patterns over time, we couldn’t analyse this association.

We have now added information about recommended first line treatment regimens to the introduction section. In lines 60 to 66 we add the statement “In Portugal the first line regimen recommends preferably the use of an Integrase Strand Inhibitor (INSTI), such as Dolutegravir (DTG), Raltegravir (RAL) or Evitelgravir (EVG/c), together with a combination of Nucleoside Reverse Transcriptase Inhibitors (NRTI), such as Tenofovir/Entricitabine (TDF/FTC), Abacavir/Lamivudine (ABC/3TC) or Tenofovir/Entricitabine (TDF/FTC). Another option is to use a Non-Nucleoside Reverse Transcriptase Inhibitor (NNRTI), such as Rilpivirine (RPV), together with a combination of NRTI inhibitors (TDF/FTC or ABC/3TC)”.

“The decline in NNRTI use has led to a drop in NNRTI resistance”

“DRV is the first-line PI of choice in many countries, and LPV is not widely used in Europe any more. PI mutations reflect these changes”

“Similarly, the decline in TAMs is consistent with a dramatic decline in the use of thymidine analogues such as AZT, ddI, d4T etc.”

We thank the reviewer for this comment. However, we don’t completely agree. While the correlation between treatment regimens’ usage and emergence of drug resistance is true; it does not correlate with persistence of DRMs, which involves a much more complex interplay between usage of ARVs, viral fitness, transmission dynamics and persistence of DRMs.

“It is not clear that the analysis only looked at the first resistance test per patient. Please clarify in the study population section of both the methods and the results. In the latter, patients are referred to, but it is the test that is being examined. This has implications when discussing time trends vis-à-vis demographic and subtype changes.”

Thank you for this comment. We have now added this information to the Methods section. In lines 92-96 we add the statement: “Viral sequences were considered to originate from ART-NP when the first ART regimen was started after or simultaneously as the sample collection for resistance testing. Only the first HIV genotypic resistance test per patient was considered for the estimation of TDR for ART-NP. Also, only the first resistance test after the first virologic therapy failure was used for the estimation of ADR in ART-EP.”

“There is no information about the time between the resistance test, diagnosis and likely time of infection. The persistence or otherwise of resistance mutations and the VL at testing will be dependent upon these factors that are unlikely to be similar between cohorts.”

We agree with the reviewer on the importance of the information about the time between the resistance test, diagnosis and likely time of infection, however, unfortunately in this database we don´t have access to this information. This is also what happens with many other drug resistance databases worldwide. These databases are frequently put together in the context of lab drug resistance testing, where clinical information such as date of diagnosis and likely time of infection is frequently missing.

However, our findings on the relationship between the presence of specific drug resistance mutations and VL, for which sample collection is performed simultaneously, will not be affected by the lack of this information.

“Has there been a change in the patterns of resistance testing over time? Has it always been the case that new diagnoses are given a genotypic test as soon as possible? In previous years, it was often not performed at diagnosis, but pre-treatment, and this could be many months or years after the initial positive test. If the time between diagnosis and resistance test has shortened over the years, this may account for higher rates of TDR detection.”

We would like to thank the reviewer for the comment. Yes, it is true that there have been changes in the patterns of resistance testing over time. However, in Portugal the genotyping resistance testing upon diagnosis is recommended since 2003, so the genotypic test is performed as soon as the patient is diagnosed. This is why we don’t have available many genomic sequences of naive patients before 2003 and for this reason we consider years 2001 and 2002 in the analysis of TDR.

“In a similar vein, has there been any change over the study period in the estimated time to diagnosis (i.e. estimated time between infection and diagnosis). If this has shortened (e.g. through more frequent testing, or targeting of high-risk cohorts), then the likelihood of TDR detection is also increased.”

Thank you for your comment. As reported above, we don’t know the date of diagnosis of the patients. Indeed, in Portugal as in many other European countries, diagnosis tends to occur earlier in specific groups, such as Men who have Sex with Men (MSM). This is one of the hypotheses we raise in the discussion for the increase of TDR. However, the following sentence has also been added to the text: “Moreover, MSM group are most frequently tested, having an earlier diagnosis, which may imply less reversion of mutations, increasing TDR detection.” was added to the discussion in lines 297-299.

Line 85 – It is stated that “Phenotypic resistance…was evaluated”. This would imply in vitrotesting of co-cultured or recombinant virus. What was done with Stanford is better expressed as “Clinical resistance…was inferred.”

Thank you for your suggestion. The sentence was changed:

Line 100-101: “Clinical resistance to ARV drugs was inferred”

The following points relate to the Results section (lines 96-252)

“This paragraph starting at line 100 has too many verbose statistics and is effectively unreadable. It would be preferable for Table S1 to be in the main text instead.”

Thank you for your suggestion. The paragraph was rewritten and Table 1 was added to the Results section.

Table 1 – The Odds Ratios are without meaning here as what they relate to is not mentioned. And the trend p-values are meaningless without stating the trend itself. Are the frequencies increasing or decreasing over time? By how much? What statistic is the p-value related to? The slope?

Thank you for your comment. To calculate the p-trend values we performed a logistic regression where the dependent variable was drug resistance (e.g. to any DRM, or to a single class or to combinations of antiretroviral classes) over a time period. This analysis gave us the Odds Ratio, which indicates the variation of frequency for each variable. This informed us about the increase or decrease of the frequencies over time. The p-trend values are associated with an odds ratio. For an OR>1.0 there is an increasing trend and for OR<1.0 there is a decreasing trend. For example, for the variable TDR, we obtained an OR=1.046 which means that for each additional year, the probability of being TDR increased 4.6%. For ADR an OR=0.867, means that, for each additional year, the probability of being ADR decreased (1- 0.867) =13.3%.

Also, how can there be 48 patients with dual-class TDR, but 47, 35, and 107 with the three 2-member combinations of PI, NRTI and NNRTI TDR? That adds up to 189. Which seems very large. Even if the 17 triple-class TDRs were counted in each two-class statistic, this still doesn’t account for the data. The same problem occurs with ADR – 1,029 patients with dual-class resistance but 1,386 with NRTI & NNRTI resistance alone. This must simply be incorrect. The remaining analysis is fatally compromised by the data being unreliable.

We thank the reviewer for highlighting this problem. All the analysis was repeated and confirmed. Table 2 was corrected as well as the text for these errors.

“Lines 131-139 – This trend analysis for TDR feels incorrect. Whilst there appears to be a trend for steadily increasing Total, NRTI and NNRTI TDR, I would be very surprised to learn that the trends were significant, let alone to the degree stated. For the area of the chart with the steepest upward slopes (2015-7) to then have an insignificant trend seems odd to say the least. The ADR data looks more trend-like, but still not very strong. Please could the authors explain in more detail how the trend p-statistics were derived, together with some sample data? This applies to a number of analyses further down the text”.

Based on this comment, we have revised our analyses. Now we only have, besides the entire study period, the time period between 2014 and 2017. We apologize but there was an error in the script that we used to calculate the logistic regression for the trend. The script was reviewed and corrected and the analysis was performed again. Indeed, we found that the odds ratio values have increased compared to the previously reported values. Table 2 and Table S1 were changed accordingly. However, for the time period 2014-2017, there are some variables that remain without significance (p>0.05), given that this is a narrow time period with fewer data points. As such, the frequency of the observations is lower which can explain the lower power of the results.

“Lines 147-150 – Why are the time intervals set at 2011-7 in the first analysis and 2014-7 in the second? These seem arbitrary when data since 2003 is available.”

Thank you for your comment. The time intervals were chosen according to different criteria. We used firstly the entire studied period (2001-2017) to do the analysis, however for naïve patients we used a shorter period (2003-2017), because we have barely any genomic sequences of naive patients before 2003 for the reasons stated above. The other time periods were chosen according to the beginning of treatment for all recommendations (2014) and, for 2011, because this corresponded to a midpoint in the analysis period. However, we acknowledge the reviewers’ comment and so that there is no further confusion, we decided to do the analysis only for 2 periods of times. Firstly, we analyzed the entire period, considering for naives the period between 2003 and 2017 and for treated patients the period between 2001 and 2017. Secondly, we analysed the time period between 2014 and 2017, as the international Antiviral Society- USA Panel recommended the treatment for all since 2014 (JAMA, 2014).

“Lines 151-60 – The inclusion of E138A in TDR is misleading as is a common polymorphism in non-B subtypes and not indicative of transmitted resistance as a public health concern. Rather it only reinforces the utility of genotyping at diagnosis. It is far from certain that it would be included in a revised surveillance mutation list. 138K is not considered polymorphic, and re-analysing the trends when including both 138A & K is hence not useful as it does not accurately reflect TDR. Reference 31 has the same issue, and was explicit in its analysis. The relevance of E138K as a TDR would only be in the context of significant historical RPV use. Only if this is the case in Portugal would this section be merited. (Certainly in the ADR section it is of interest notwithstanding.)”

Thank you for your comment. Since E138A and E138K may have different effects in terms of resistance, we perform the calculations again considering E138A and E138K separately.

When considering E138K mutation, TDR increased from 9.4% (95%CI: 8.8-10.1) to 9.5% (95%CI: 8.9-1.0%), while TDR to NNRTIs increased from 5.0% (95%CI: 4.5-5.5%) to 5.1% (95%CI: 4.6-5.7%). However, when E138A polymorphism was analysed separately, TDR increased to 11.9% (95%CI: 11.3-12.8 %) and TDR to NNRTIs increases to 7.6% (95%CI: 7.0-8.2%).  According to Stanford, E138K is a nonpolymorphic mutation that is selected in a high proportion of patients receiving RPV and that reduces RPV susceptibility 2 to 3-fold; on the other hand, E138A is a polymorphic mutation that ranges in prevalence from about 1% to 5% depending on the subtype. While it can be a polymorphic position in specific subtypes, it is important to consider that it reduces Rilpivirine (RPV) susceptibility about 2-fold. According to the RPV package insert, the presence of E138A prior to therapy may reduce the antiviral activity of RPV.

Furthermore, since a) current Portuguese guidelines recommend the use of RPV as first line therapy in combination with two NRTIs in patients that present a viral load <100000copies/ml; and b) the Portuguese epidemic presents a high percentage of non-B subtypes, we think that is a very important polymorphic position to take into account for the surveillance of TDR in Portugal.

The paragraph on results (lines 180-188) about E138A/K was changed accordingly:

 “When considering E138K mutation, TDR increases from 9.4% (95%CI: 8.8-10.1) to 9.5% (95%CI: 8.9-1.0%), while NNRTIs increases from 5.0% (95%CI: 4.5-5.5%) to 5.1% (95%CI: 4.6-5.7%). However, when E138A polymorphism is separately analysed TDR increases to 11.9% (95%CI: 11.3-12.8) and NNRTIs increases to 7.6% (95%CI: 7.0-8.2%). According to standford, E138A is a polymorphic mutation that ranges in prevalence from about 1% to 5% depending on subtype and it reduces RPV susceptibility about 2-fold. Since current Portuguese guideline recommends the use of Rilpivirine as the preferential NNRTI to use combined with two NRTI as first line regimen, when the patient presented a viral load <100000copies/ml, we think that is important to do this analysis in the Portuguese context.”

“Lines 171-5 – It makes no sense to talk of the 2003-17 cohort as a single entity here, given the emphasis on time trends. Similarly for the three classes.”

We agree with the reviewer. We could have considered other time periods, for example the period between 2006-2017 where the increasing trend seems to initiate. However, given our initial hypothesis of the impact of treatment for all recommendation on the increase of TDR, as well as the clearly linear increasing trend for TDR after that, we decided to consider the 2014-2017 period for the second statistical analysis.

“Lines 176-187 – These trend analyses suffer from the same problems as the TDR ones. For example, the p-value for NNRTI detection is asserted to be <0.001 despite the change being minimal and the chart plot being essentially flat. The problems with the numbers in the table are already stated, and these data cannot be considered reliable. Trend analysis on the post-Treatment for All period is pointless given its brevity, and certainly cannot be meaningfully compared to the longer time trend. Multi-class analysis is similarly unsuitable.”

For TDR, the analysis was significant both in the whole time period, as well as in the 2014-2017 time period. This is an important finding given our initial hypotheses of increase of TDR due to Treatment for All. Our analysis indicated a 4.6% increase per year of TDR in the 2003-2017 compared to 16.9% increase of TDR in the 2014-2017. These results are significant despite the lower amount of data analysed in this period: for example for NRTI resistance OR= 1.337; chance of being diagnosed with TDR 33.7% increase per year; p=0.001 and for NNRTI OR=1.132; chance of being diagnosed with TDR 13.2% increase per year; p=0.044. As such, we consider important to take into account this time period.

“Lines 192-8 – It seems that K103N and M184V are increasing in the drug-naïve cohort concomitant with a sharp decline in the experienced cohort. This needs exploring. As with other trend analyses, I am sceptical of the p-values associated with the individual mutations, particularly as there is no raw data to view.”

We thank the reviewer for this comment, given the importance of this finding. K103N and M184V are increasing in the drug-naïve cohort, as shown in the graph. However, we went back to our database and found that the graph of treated patients had a small mistake in the count of K103N and M184V in the last 3 years. This was already corrected and the graph presented in figure 2 has been changed. We observed a small variation of these mutations over the years instead of the decrease previously shown.

Table S5 was added at supplementary file containing all mutations analysed, included the odds ratio and p for trends over years for mutations frequencies. The logistic regression for all mutations was also corrected.

“Lines 224-9 – If true, the reported frequencies of acquired INSTI resistance are alarmingly and astonishingly large. Some information on the selection of these samples is essential, as it must be the case that these sequences derive solely from patients failing INSTI regimes.”

We thank the reviewer for highlighting this point. When we referred to “treated patients “, in this case we wanted to refer to patients treated with different classes of antiretrovirals but who had virologic failure for INSTIs. We changed in these sentences (lines 257-258) “patients treated” to “patients presenting virologic failure to INSTIs”

Lines 231-43 – This area needs work:

  • “Given the multivariate analysis, there seems to be no value in the univariate results. The significance of e.g. sex/subtype/origin/age at diagnosis etc. cannot be easily disentangled from each other.”

The univariate results were presented in the supplementary file in Table S2 and S3, showing the significant variables as well as the non-significant variables. As we explain in the supplementary file, the univariate analysis of factors associated with HIV drug resistance included the following covariates: gender, age at diagnosis, risk groups, geographic region of origin, subtype, CD4 value at diagnosis and viral load. Factors significantly associated with TDR in the univariate analysis (p<0.05), factors with a p value lower than 0.2 and factors considered with biologic value (gender and age at diagnosis) were included in the multivariate logistic regression (showed in Table 2). As such, given the univariate analysis guides the selection of factors to be considered in the logistic regression, and that these results are only presented in the supplementary file, we find it important to keep the results presented in the manuscript supplement.

  • “Why are so few of the factors in Table S3 present in Table 2? They should all be included, if only to show non-significance.”

This is because in table 2 we only present factors included in the logistic regression, which were selected based on the univariate analyses results, as has been explained in the previous comment.

  • “The data on the individual mutations are not present in Table 2 but are referred to in the text.”

We thank the reviewer for highlighting this. Table S5, which includes the analyses of individual mutations, was added in the Supplement.

  • “Have the analyses been corrected for multiple comparisons (e.g. Holm-Bonferroni?)”

There is a misunderstanding concern multiple comparison because we performed univariate and multivariate logistic regression analyses and we believe that Holm-Bonferroni might be used if we intend compare median and mean between groups (analysis of variance).

  • “Replace “Gender” with “Sex” in Tables 2, S3 & S4.”

Thank you for the suggestion. Changes were made.  

  • “Table S2 – How the OR and p­-values were obtained needs much more transparency.

As we have previously explained, the OR and p-values were obtained with logistic regression where the dependent variable was drug resistance (e.g. to any DRM, or to a single class or to combinations of antiretroviral classes) over a time period.

  • The choice of time intervals looks like p-mining. Why are there separate statistics for Sep 2014-2017 and for 2015-7?”

As previously mentioned we decided to keep only 2 periods of time in the text. the period from 2001 (ADR) or 2003 (TDR) to 2017 and from 2014 to 2017 (after Treatment for All recommendations started). The necessary changes were made in the manuscript and in the supplementary file.

Finally, there are issues with the Discussion. By focussing on demographic trends, the consequences of the likely temporal trends in testing and treatment mentioned above are overlooked.

We thank the reviewer for this comment. The objective of the paper was to analyse the trends of ARV drug resistance, both acquired and transmitted, across time, and when possible, to understand an hypothesize on why changes occur. While changes in testing patterns and on treatment usage have changed across time, it is difficult to quantify those changes and, based on the information we have in this database, we can’t gather that information. However, this is a very interesting topic which we plan to analyse in the future; but to be able to do that we will need to collect other databases. While for treatment it is possible to get that information from paid sources; for testing the analysis seems to be a bit more complex, as it implies knowing the time that spanned between infection and diagnosis, which is not so easy to estimate.

“Lines 257-8 – Please expand upon why TDR would be correlated with the Treatment for All recommendations? Aside from a temporal coincidence, what is the suggested mechanism by which the latter influenced the former? Increased testing and earlier detection? Refer to the Methods critique above to make sure all these putative causes are mentioned.”

The increased use of antiretroviral therapy is associated with external selective drug pressure acting on the virus.  The more patients we put on treatment, the more drug selective pressure is acting on the virus. On the other hand, we are putting more patients on treatment in countries where adherence is reportedly very low. As such, as we treat more patients, we expect to see more drug resistance emerging, which can then propagate through TDR.

“Lines 264-9 – Is the statement about increasing sub-Saharan migration consistent with the one about increasing MSM? Are both populations increasingly represented in the Portuguese HIV cohort, or is this an artefact of changing testing regimes over time? How can the sub-Saharan migrant population both have lower TDRs, as evidenced by the non-B rates in Table 2 (significantly lower for all but NNRTI), and simultaneously contribute to increased rates of TDR?

Indeed, these are two concurrent hypotheses, which can however be addictive. In fact, we can see both effects independently: increasing TDR in MSMs and increasing TDR in migrants. These populations are both an important part of the HIV-1 Portuguese epidemic, accounting for a high number of cases each year. We have tried to compare the trends of TDR across time in male vs female (using male as a surrogate for MSM) and migrants vs non-migrants. However, the size of the samples does not allow to make any firm conclusion; although there seems to be a more evident increase of TDR in migrants, as reinforced by our recent publication “Pimentel V, Pingarilho M, Alves D, et al. Molecular Epidemiology of HIV-1 Infected Migrants Followed up in Portugal: Trends between 2001-2017. Viruses. 2020;12(3).” where we show a significant increase of TDR in migrants. Concerning the B vs non-B comparison, while we see lower TDR rates in non-B subtypes, this does not mean that TDR is not increasing in non-B subtypes. In fact, recent reports indicate increasing levels of TDR in Africa, where non-B subtype prevail.

How does the hypothesized impact of the rapid onward transmission suggested for the MSM cohort compare to the findings alluded to in lines 293-5 where TDR was found throughout a late-presenting cohort and associated with clusters of transmission?

Another ongoing study at our lab, that compared Late Presenters (LP) with Non-Late Presenters (NLP) showed that TDR was higher in the NLP group, which are more frequently part of transmission clusters. This study also showed us that NLP group is composed mainly of MSM.

We have adapted the text (lines 331-333):

“Another ongoing study at our lab, comparing a late presenters cohort with non-late presenters, interestingly showed that individuals involved in transmission clusters were more frequently non-late presenters and presented more frequently TDR 29. This suggests that mutations associated with resistance are maintained and transmitted forward in these clusters, indicating that they don’t reduce viral transmissibility”

Why favour any of these hypotheses when the major question – “are the data a function of national treatment and testing protocols?” – is not raised? Does the inverse relationship between TDR and ADR in recent years point in one particular direction when looking at all these theories?

Given that the spread of drug resistance is a global problem and that about 25% of yearly infections are detected in migrants, we should certainly not say that our results are a function of national treatment protocols. We would better say that it can be influenced both by treatment and testing protocols. In fact, the fact that ADR is significantly decreasing is certainty a result of the usage of antiretroviral regimens with a higher genetic barrier. However, for TDR, we know that if we diagnose earlier, we are more likely to detect it. This could indeed be the case and we have added that sentence to the discussion:

Lines 297-299: “Moreover, MSMs are more frequently tested, having an earlier diagnosis, which may imply less reversion of mutations, increasing TDR detection. “

“Lines 270-1 – E138A does not impact upon EFV or NVP, and K only weakly. The effects stated are for K103NS.”

Given this and other comments on E138A/K, we have redone the analysis.  

Lines 180-188: “When considering E138K mutation, TDR increases from 9.4% (95%CI: 8.8-10.1) to 9.5% (95%CI: 8.9-1.0%), while NNRTIs increases from 5.0% (95%CI: 4.5-5.5%) to 5.1% (95%CI: 4.6-5.7%). However, when E138A polymorphism is separately analysed TDR increases to 11.9% (95%CI: 11.3-12.8) and NNRTIs increases to 7.6% (95%CI: 7.0-8.2%). According to Stanford, E138K is a nonpolymorphic mutation that is selected in a high proportion of patients receiving RPV and that reduces RPV susceptibility 2 to 3-fold; on the other hand, E138A is a polymorphic mutation that ranges in prevalence from about 1% to 5% depending on the subtype. While it can be a polymorphic position in specific subtypes, it is important to consider that it reduces Rilpivirine (RPV) susceptibility about 2-fold.”

Line 318-320: “E138A/K was also considered separately, since it confers resistance to Rilpivirine, which is included in the preferential regimens used in Portugal.”

“Line 273 – M184V does not cause high-level resistance to abacavir. Again, the syntax of the sentence is misleading.”

We apologize for this error. We have changed the text to:

Line 314-315:” and M184VI which causes high-level resistance to first-line recommended drugs 3TC and FTC 22

“Lines 274-5 & line 331 – L90M causes clinically significant PI resistance even in the absence of further mutations.”

Yes, it’s true according to Stanford L90M can cause clinically significant PI resistance alone, so we changed the sentence mentioned above.

The following sentence (Line 315-316): “L90M was the resistance mutation to PIs with highest prevalence. It causes reduced susceptibility to ATV and LPV when in combination with other PI-resistance mutations.”  was changed to “L90M was the resistance mutation to PIs with highest prevalence and it causes reduced susceptibility to ATV and LPV “and the other one (Line 369) “…and L90M reducing susceptibility in combination with other PIs mutations, specifically to ATV and LPV.” was changed to “and L90M reducing susceptibility, specifically to ATV and LPV”.

Lines 275-6 – Please remove "not that worrying" as it still precludes NNRTIs for the lifetime of the patient, even if it is not a first-line drug in 2020.

It was removed.

“Lines 278-84 – I would downplay the E138AK results as they are of limited information given the background frequency of 138A in non-B subtypes. Unless the analysis ignored 138A and only explored 138K, their impact on the analyses should be minimised. As above, there is no suggestion that 138A would be in a revised SDRM list (although there is a stronger case for 138K).”

Please see comment above concerning codon 138.

“Lines 285-92 – The impact of subtype on TDR prevalence as shown in Table 2 needs to be reconciled with this paragraph (see also lines 264-9).”

Despite the fact that TDR is lower in non-B subtypes, our data indicates that TDR is increasing both in subtype B and in non-B subtypes.

“Lines 299-303 – That E138A occurs as a natural polymorphism indicates it has minimal fitness impact, if any. Furthermore, how do the authors account for L being wild-type at position 90 of protease, if M confers “significantly higher fitness”? Either it doesn’t, and the referenced paper is mistaken, or there is a very odd phenomenon occurring here.”

As we already explained, the analysis was now performed separately for E138A and E138K, but we think that is important to consider this polymorphism in drug resistance surveillance, since this polymorphism does confer reduced susceptibility to Rilpivirine, which is included in the preferential regimens used in Portugal.

In what concerns to L90M, we are referring to transmission fitness and it is known that relation between L90M and transmission fitness has been demonstrated in different papers (for example  Kunhert et  al, PLoS Pathog. 2018 Feb; 14(2): e1006895, reported that L90M mutation in the protease gene was found to have significantly higher fitness than the drug sensitive strain). They also reported that “mutations associated with resistance to reverse transcriptase inhibitors were found to be less fit than the sensitive strains: 67N, 70R, 184V, 219Q”, which is also in accordance with what we have discussed in the text and with the findings reported in other cited papers.

“Lines 304-6 – I cannot see in the Results section the data supporting these claims about “significantly lower levels of VL” associated with the three mutations. Also, lower than what? Drug-naïve patients without these mutations? Individually or en masse?”

This statement is at the population level. We apologize for not including this data in the previous version. Table S4, which includes this information, was already added to supplementary file and “significantly lower levels of VL” (lines 343-344) was changed to “TDR mutations presented statistically significant lower levels of VL than ART-NP without DRMs”.

“Lines 304-317 – It is a major leap to go from an observed lower VL with certain mutations (that are in turn associated with other demographic and temporal characteristics) to discuss transmissibility, as if the latter was purely a function of VL at the moment of genotyping (irrespective of time since infection) and that the VL is substantially affected by these mutations directly. There is no data on the transmitting patients to support these claims. Phylogenetic methods can only partly mitigate this missing information. This paragraph needs to be looked at again, and probably downplayed considerably.”

This paragraph is a mere hypothesis to support future studies and it is only speculative. We do mention in the text that transmissibility of HIV strains should be determined not only by the VL, but by a trade-off with other aspects such as infectiousness and disease progression and that our findings should guide future directed studies to investigate an eventual association between VL and transmissibility. While setpoint VL could be considered much more important, we do find relevant that strains presenting these mutations present with VL at the time of drug resistance testing.

We changed the last sentence of the paragraph to (lines 353-355):

“Our consistent finding of lower VLs in patients carrying M184V and K103N, despite increasing prevalence, warrants future directed investigations of these results.”

“Line 320 – How can a multi-centre European cohort be in Switzerland?!”

This was a typo error, which has been changed to “in a multi-center European cohort (1997-2008) in Switzerland” for “in a multi-center cohort (1997-2008) in Switzerland”.

 “Line 318-327 – Given the dramatically improved suppression viral rates of ARV regimes over the last five years or so, together with better testing systems, just where is the TDR coming from? If it is asserted that it is arising from sub-Saharan migrants, then there needs to be more finely-resolved data presentation in the Results section around this.”

We have published another manuscript where we look specifically at migrants and where the study is specifically designed to look at this hypothesis. This manuscript is cited in the text: “Pimentel V, Pingarilho M, Alves D, et al. Molecular Epidemiology of HIV-1 Infected Migrants Followed up in Portugal: Trends between 2001-2017. Viruses. 2020;12(3).”

“Lines 328-334 – The hypotheses here need to be related to those made regarding several of these mutations and viral fitness (lines 293-317 roughly). Either way, the reduction in TAMs is almost certainly a consequence of prescribing patterns being radically different across the timeframe discussed. Thymidine analogues are very rarely prescribed in Europe these days, although they are still widely used in LMICs.”

We thank the reviewer for highlighting this point. We don´t have data available to do this type of study, however we added the following sentence to the discussion: “Moreover, antiretroviral regimens changed over time, with a decrease in the prescription of thymidine analogues in Europe, which could have implied the observed decrease of TAMs.”

“335-345 – Firstly, can the relationship between male sex and subtype B virus truly be disentangled? Seeing as >85% of diagnoses have no transmission route, then essentially these are unlinked in the analyses. Secondly, does this not simply reflect time since infection? Most of these MSM subtype B viruses will have a long treatment history. Given the lack of clarity over the sampling, this remains unclear.”

Yes, it is true that this reflects time of infection, since subtype B is being treated a long time ago.

We have added this sentence to the discussion:

Lines 384-386: “Finally, subtype B which caused the initial pandemic in the Western World, has been subject to selective pressure by ARV drugs for much longer than other subtypes, and this should explain why we find significantly more ADR in this subtype.”

The following points are minor points of composition:

  • Consider replacing DN / TP with treatment-naive / treatment-experienced, or ART-naive / ART-experienced. This has been done in Table S1 and should be used throughout

We replaced all drug- naïve patients for ART- naïve patients (ART-NP) and treated patients for ART-Experienced patients (ART-EP).

  • Express p-values as <10-xwhen it is below 0.001 - having a p-value of 0.000 is not appropriate. However, the trend analysis needs revisiting.

All the values that were p=0.000 were changed to p<0.001. We maintained the format of p<0.001 instead of <10-x in order to keep all p values with the same format and make it easier to visualize

  • Lines 27, 133, 177, 290 – Capitalise or quote “Treatment for All” as per line 258.

Have all been changed.

  • Line 52 - expand ARV & ART at first use of the abbreviation.

We have changed this accordingly.

  • Line 98 - remove the second instance of "patients"

We have removed it.

  • Lines 103 or 107 - MSM used (correctly) on line 248 but not here. Please revise.

The expression “sex between men” was changed by “homosexual contact”.

  • Lines 146, 155, 167, 332 - "over time" instead of "overtime"

We have changed it accordingly.

  • Figure 2 – Several points:
    • The y-axes of 2A & 2B are incomplete – only half of the mutations are present.

We have changed it accordingly.

  • The legends and axis titles need to be consistent (NRTIs vs. NRTI in 2A & 2B, and for 2C & 2D, patients vs. patient, and the order of mutations in the legends).

We have changed it accordingly.

  • Remove the decimal place from the x-axis in 2B and from the y-axes of 2C & 2D.

We have changed it accordingly.

  • Figure 3 - Remove the decimal place from the x-axes.

We have changed it accordingly.

  • Lines 223-4 – There is a fullstop where the sentence should continue “…not yet recommended by health authorities.”

We have changed it accordingly.

  • Line 227 – Replace “intermedium” with “intermediate”

We have changed it accordingly.

  • Line 293 – This sentence needs tidying up.

The sentence:” Another interesting finding was that another ongoing study at our lab, enrolling a late presenters’ cohort with primary resistance, showed that the prevalence of TDR was higher in individuals that were involved in transmission clusters” was rewritten to “Another ongoing study at our lab, comparing a late presenters cohort with non-late presenters, interestingly showed that individuals involved in transmission clusters were more frequently non-late presenters and presented more frequently TDR29”.

  • Line 298 – “has also shown” for “have also shown”

We have changed it accordingly.

  • Lines 300-3 – There is another rogue fullstop, here between ref 33 and only […190A) 33, only the…]

We have changed it accordingly.

  • Line 332 – Too many “important”s.

We have changed it accordingly.

  • Table S1 – Spelling for Sub-Saharan

We have corrected it.

Finally, the authors should not place the Tables and Figures inline with the text as it compromises the readability. It is better practice to collate them at the end of the manuscript.

The Tables and Figures were all at the end of the manuscript, however the editor placed it in the middle of the text.

Reviewer 2 Report

In this manuscript, Pingarilho and collaborators analyze the trend of drug resistance prevalence in Portugal in between 2001 and 2017. This study is relevant for the implementation of more effective treatment guidelines. However, I find a few shortcomings in this study that are outlined below.

Why the 2018-2020 period is not analyzed? I think this period is relevant and including it in the study would be interesting.

It is not clear how the genomic data is obtained. Sanger sequencing? Deep sequencing techniques? And, which is the frequency of the reported mutations in each patients? If the reported mutations represent the main sequence present in each individual then I assume the treated patients are failing therapy. A mean prevalence of ADR of 69% seems too high if those resistant mutants are dominant. It is also not clear how these patients are selected and why they have had an HIV-1 drug resistance test. This information is important to know whether a bias is introduced in the selection of the study population.

What is the time since infection for drug naïve patients? Resistant mutations are frequently associated with a fitness cost, so one would expect that the occurrence of these TDR mutations diminish over time in an untreated patient. Are there known the sequences before starting treatment in treated patients? If they are not known, then some of the ADR mutations could be TDR mutations that have been fixed after ARV treatment.

For most of the population the mode of transmission is unknown (table S1). I assume that their risk group is also unknown. In addition, the country of origin is unknown for a large percentage of patients (table S1). Thus, conclusions about the association of these parameters with ADR or TDR  should be taken with care.

Lines 145-150: The data in this paragraph refers to table 1 and S2. However, I think there is some information missed in table S2 (trends in TDR). I do not see the percentages of each class mutation over time and the periods reported in the table do not coincide with those indicated in the text. In addition, in line 147 after pfortrend=0.668 it should be added for triplicates.

Lines 151-158: There is no reference to any figure or table for this paragraph (data not shown?)

Lines 161-163: these mutating are supposed to be reported in Figure 2A. However, I do not see some of them in this figure. Actually, some of the bars in the chart of this figure (and in figure 2B) do not have a label and some of the labels are not clearly assigned to one bar.

Lines 182-187: some of the data included in this paragraph in not reported in Table 1 and table S2.

Table S2: This table reports the period from September 2015 to 2017 for ADR. However, the headline of the table indicates trends of ADR between 2001-2017.

Lines 188-191: same comment than for lines 161-163.

Line 193: The percentages reported in figures 2C and D are higher.

Lines 222-229: In which figure or table are reported these data?

Lines 235-238 and lines 247-250: As mentioned above for most of the study population the risk group and country of origin is unknown. Thus, conclusions reached regarding the significance of their association with TDR or ADR should be evaluated with care.

Lines 241-243: The specific mutations mentioned in this sentence are not individually reported in Table 2.

The quality/resolution of the figures should be improved. The font size in some figures is too small and are difficult to read.

Minor issues:

Line 73: It would be useful to include a brief description of RegaDB

Line 98: What does it mean that “patients presented viral sequences”? What happen with the sequences of the other 880 patients?

Line 223: “recommended. By”

References to figures and tables is usually done at the end of each paragraph, which sometimes results confusing. As an example, in the description of the characteristics of the Portuguese population (lines 100-116) I would include an introductory sentence at the beginning of the paragraph indicating that the population characteristics are described in table S1, instead of indicating it at the end of the paragraph. In other cases, when in one paragraph there are data related to more than one figure or table, the figure or table could be indicated after each sentence.

Author Response

To the Board of editors.

We have recently submitted our manuscript entitled “Increasing prevalence of HIV-1 Transmitted Drug Resistance in Portugal: implications for first line treatment recommendations” to Viruses. We would like to acknowledge all the comments and suggestions provided by the journal editor and by the reviewers. We are now submitting the revised version, where we address all the comments and suggestions provided by the reviewers and by the journal editor. Overall, we feel that revision process has greatly contributed to a higher quality paper. A point-by-point answer is provided below. We hope to have answered the journal requirements and reviewers’ concerns and to provide a new version of the manuscript suitable for publication in Viruses.

Sincerely yours,

Marta Pingarilho

Reviewer 2

“Why the 2018-2020 period is not analyzed? I think this period is relevant and including it in the study would be interesting”

We thank the reviewer for highlighting this point, however we cannot add this information in this publication.

This article was carried out in the context of a project that ended at the end of 2017 and we only had authorization from the hospital ethics committee to use data until the end of 2017. However, we hope that soon we will have access to the data from 2018 to 2020 and do a small study on the last years.

“It is not clear how the genomic data is obtained. Sanger sequencing? Deep sequencing techniques? And, which is the frequency of the reported mutations in each patients? If the reported mutations represent the main sequence present in each individual then I assume the treated patients are failing therapy. A mean prevalence of ADR of 69% seems too high if those resistant mutants are dominant. It is also not clear how these patients are selected and why they have had an HIV-1 drug resistance test. This information is important to know whether a bias is introduced in the selection of the study population.”

We would like to thank the reviewer for the comment. In the methods section the second paragraph that corresponds to “Drug resistance analyses and subtyping” has been improved and the following paragraph was rewritten (lines 90-96):

“The genomic data included protease and reverse transcriptase sequences obtained through population sequencing completed at Laboratório de Biologia Molecular of Hospital de Egas Moniz/CHLO. Viral sequences were considered to originate from ART-NP when the first ART regimen was started after or simultaneously as the sample collection for resistance testing. Only the first HIV genotypic resistance test per patient was considered for the estimation of TDR for ART-NP. Also, only the first resistance test after the first virologic therapy failure was used for the estimation of ADR in ART-EP.”

“What is the time since infection for drug naïve patients? Resistant mutations are frequently associated with a fitness cost, so one would expect that the occurrence of these TDR mutations diminish over time in an untreated patient. Are there known the sequences before starting treatment in treated patients? If they are not known, then some of the ADR mutations could be TDR mutations that have been fixed after ARV treatment.”

Thank you for your comment. Unfortunately, we don’t have information about infection date available in the database, so we cannot know the time since infection for drug-naïve patients. We understand the importance of this issue, which we are trying to explore in another analysis we are performing with a different cohort. Concerning the paring of sequences before and after starting treatment, this is something we might be able to evaluate in future studies and we thank the reviewer for bringing this up. However, this analysis on its own will make a complete new study.

“For most of the population the mode of transmission is unknown (table S1). I assume that their risk group is also unknown. In addition, the country of origin is unknown for a large percentage of patients (table S1). Thus, conclusions about the association of these parameters with ADR or TDR should be taken with care.”

Thank you for this note. We based our conclusions on the available data. Any other associations we are now careful to show that are only speculative and hypothetical and based on the comparison of our results with other available studies.

“Lines 145-150: The data in this paragraph refers to table 1 and S2. However, I think there is some information missed in table S2 (trends in TDR). I do not see the percentages of each class mutation over time and the periods reported in the table do not coincide with those indicated in the text. In addition, in line 147 after pfortrend=0.668 it should be added for triplicates.”

We thank the reviewer for highlighting these points and we apologize for the lack of some results in the initial version.

We have now added Table S5 to the Supplement, which reports the percentages of each class mutation over time.

Furthermore, we now clarify the time intervals analyzed. For ADR, we used the period 2001-2017 for the analysis. However, for naïve patients we used a shorter period (2003-2017), since in Portugal the recommendation to test for drug resistance after diagnosis started only in 2003. The other time periods were chosen according to the beginning of treatment for all recommendations (2014) and, for 2011, because this corresponded to a midpoint in the analysis period. We have also only performed the analyses considering two time periods. First, we considered the entire period: for naives between 2003 and 2017 and 2001 and 2017 for treated patients as specified above. Moreover, we analysed the time period between 2014 and 2017, which corresponds to the beginning of the treatment for all recommendations (Antiretroviral Treatment of Adult HIV Infection: 2014 Recommendations of the International Antiviral Society–USA Panel; Clinical Pharmacy and Pharmacology; JAMA; JAMA Network).

“Lines 151-158: There is no reference to any figure or table for this paragraph (data not shown?)”

As suggested by another reviewer, we have reanalysed the data to account separately for E138A and E138K. We have changed the text accordingly. As these mutations are not part of the surveillance drug resistance mutations (SDRMs) list, we added these percentages only to the text.

“Lines 161-163: these mutating are supposed to be reported in Figure 2A. However, I do not see some of them in this figure. Actually, some of the bars in the chart of this figure (and in figure 2B) do not have a label and some of the labels are not clearly assigned to one bar.”

Thank you for highlighting this. The figures have been corrected.

“Lines 182-187: some of the data included in this paragraph in not reported in Table 1 and table S2.”

Thank you for your comment. Table S6 was added to supplementary file.

“Table S2: This table reports the period from September 2015 to 2017 for ADR. However, the headline of the table indicates trends of ADR between 2001-2017.”

The time periods on this table were changed as well as the legend of the table.

“Lines 188-191: same comment than for lines 161-163.”

Thank you for highlighting this. The figures have been corrected.

“Line 193: The percentages reported in figures 2C and D are higher.”

Thank you for highlighting this. We apologize but there was an error. The sentence was changed to “when mutations had a prevalence greater than 0.1% for ART-NP and 1.0% for ART-TP”.

“Lines 222-229: In which figure or table are reported these data?”

These data are not shown. We added this information to the end of the paragraph.

“Lines 235-238 and lines 247-250: As mentioned above for most of the study population the risk group and country of origin is unknown. Thus, conclusions reached regarding the significance of their association with TDR or ADR should be evaluated with care.”

We agree with this comment. However, we note that significant associations related to risk group and country of origin are only reported in the results and not further highlighted in the discussion and conclusions.

“Lines 241-243: The specific mutations mentioned in this sentence are not individually reported in Table 2.”

Thank you for your comment. Table S4 was already added to supplementary file and to this paragraph.

“The quality/resolution of the figures should be improved. The font size in some figures is too small and are difficult to read.”

We have corrected this problem and everything is now more legible.

Minor issues:

“Line 73: It would be useful to include a brief description of RegaDB”.   

We have added the following sentence:

Lines 80-83: “All patients’ data were generated in the context of routine clinical care and collected in RegaDB 8.  RegaDB is a free and open source data management and analysis environment for infectious diseases, which allows clinicians to store, manage and analyse patient data, including viral genetic sequences. “

“Line 98: What does it mean that “patients presented viral sequences”? What happen with the sequences of the other 880 patients?”

The database is composed by 12792 patients (2001-2017) with some demographic characteristics, however not all had genomic sequences available, because they didn´t performed genomic resistance tests, as such we didn´t include these patients in our analysis.

Line 223: “recommended. By”

This sentence has been changed accordingly.

References to figures and tables is usually done at the end of each paragraph, which sometimes results confusing. As an example, in the description of the characteristics of the Portuguese population (lines 100-116) I would include an introductory sentence at the beginning of the paragraph indicating that the population characteristics are described in table S1, instead of indicating it at the end of the paragraph. In other cases, when in one paragraph there are data related to more than one figure or table, the figure or table could be indicated after each sentence.

Everything has been revised and changed accordingly; and highlighted in the text.

Reviewer 3 Report

Submitted study performed in twelve thousand HIV-1-infected individuals in Portugal between 2001 and 2017 shows decrease of ADR and increase of the total TDR reaching a value of 13.1% by the end of 2017. Identification of predictors of TDR plays primordial importance in selection of the first line therapy recommendation. While decreasing ADR seems to be caused mainly by the increasing efficacy of ARV therapy, interpretation of the total TDR increase is more complex. TDR seems to be mainly driven by virus determinants such as viral subtype and fitness of the virus. It correlates with the implementation of the “treat-all” recommendations and with the changing face of the pandemic, including an increasing proportion of MSMs, less reversion of DRM and the increasing presence of migrants in Portugal from countries where TDR is more prevalent. Not surprisingly, studies performed in other countries showed different results, with the TDR prevalence decreasing or stabilizing over time.

Inherent problem of this study consists in increase of the total TDR including TDR to NRTIs and NNRTIs but decrease of TDR to PIs. Also, dual and triple class resistance to NRTIs/NNRTIs/PIs combinations increased. While individuals with combined NRTIs/NNRTIs/PIs resistance are rare, number of individuals with TDR to PIs are comparable with TDR to NRTIs and NNRTIs. In addition, while NRTI resistant mutation M184V was selected to illustrate increasing trend of TDR in the last five years of the study (Fig. 2C), it seems that the whole 16-years follow-up has decreasing tendency, very similar to that of PI resistant mutation L90M. These discrepancies should be discussed and explained. Please compare also the cohorts studied in other reports on the trends of evolution of TDR showing different results.

To facilitate reading of manuscript to scientific community outside of the HIV research field, please explain meaning of “90-90-90 target” (l. 47), and indicate in Introduction the criteria for inclusion of HIV-1-infected individuals in TDR and ADR categories. Also, why TAM DRM (shown in Fig. 1) are not shown in Table 1 and are not defined in the text (l. 189).

Figure 2. Please enlarge the font size for HIV DRM in panel A and B. Explain in the figure legend why 2 histograms are shown for each DRM (panels A and B, except of K65R in the panel A). Why just M184V, K103N, and L90M were selected for presentation in panels C and D?

Minor points

line 104: Number after a period (. 67.3…..)

lines 119-120: please organize NNRTI and NRTI in the same order in the text as in Table 1

line 151: please indicate a reference for the “previous results”. Please indicate more explicitly that E138A/K mutation confers resistance to NNRTIs (l. 151) and that related calculations concern Table 1.

line 222: strand instead of stand

line 238: Please move up reference to Table S3 after the first sentence of the paragraph.

Please use decimal point in numbers. Sometimes decimal commas are used.

Author Response

To the Board of editors.

We have recently submitted our manuscript entitled “Increasing prevalence of HIV-1 Transmitted Drug Resistance in Portugal: implications for first line treatment recommendations” to Viruses. We would like to acknowledge all the comments and suggestions provided by the journal editor and by the reviewers. We are now submitting the revised version, where we address all the comments and suggestions provided by the reviewers and by the journal editor. Overall, we feel that revision process has greatly contributed to a higher quality paper. A point-by-point answer is provided below. We hope to have answered the journal requirements and reviewers’ concerns and to provide a new version of the manuscript suitable for publication in Viruses.

Sincerely yours,

Marta Pingarilho

Reviewer 3

“Inherent problem of this study consists in increase of the total TDR including TDR to NRTIs and NNRTIs but decrease of TDR to PIs. Also, dual and triple class resistance to NRTIs/NNRTIs/PIs combinations increased. While individuals with combined NRTIs/NNRTIs/PIs resistance are rare, number of individuals with TDR to PIs are comparable with TDR to NRTIs and NNRTIs. In addition, while NRTI resistant mutation M184V was selected to illustrate increasing trend of TDR in the last five years of the study (Fig. 2C), it seems that the whole 16-years follow-up has decreasing tendency, very similar to that of PI resistant mutation L90M. These discrepancies should be discussed and explained. Please compare also the cohorts studied in other reports on the trends of evolution of TDR showing different results.”

Thank you for these comments. Concerning the levels of TDR to dual or triple class combination, we don’t see how the results can be discrepant. Resistance to PIs alone is 2.8%, compared to 5% to NNRTIs and 4% to NRTIs. Furthermore, the fact that there is a decreasing trend for PIs is not incompatible with the fact that there are decreasing trends for NRTIs and NNRTIs. This could be caused by changes in drug usage patterns, as we have discussed in the text:

Lines 313-316: When looking at specific drug classes, we find an increasing trend of TDR to NRTIs and NNRTIs, but a decreasing trend of TDR to PIs. We hypothesize that this could be due to patterns of drug usage, with a infrequent use of PIs at first line in Portugal. Another explanation could be the larger flexibility of protease, which could imply a lower probability of fixation of mutations.

Concerning specific mutations, we have now complemented the information that had been provided before. We added additional information concerning the trends of DRM in the Supplement (table S5). For L90M, we see a decreasing trend which is significant. For M184IV, the trend is not significant. Indeed, we agree that there is initially an apparent decreasing trend in the first years of the study period. However, after 2012, this trend is clearly increasing. We see the same pattern for K103N, which has a significant increasing trend.

To clarify this, we have now changed the sentence:

Line 225-226: Although M184V did not presented a significant trend (pfor-trend=0.473) we observed an increasing trend for M184V since 2012 (Figure 2(C)).”

“To facilitate reading of manuscript to scientific community outside of the HIV research field, please explain meaning of “90-90-90 target” (l. 47), and indicate in Introduction the criteria for inclusion of HIV-1-infected individuals in TDR and ADR categories. Also, why TAM DRM (shown in Fig. 1) are not shown in Table 1 and are not defined in the text (l. 189).”

Thanks for pointing out this. We have now added this information to the introduction section. In lines 47-49 we rewrote the statement “In 2014, the WHO proposed the 90-90-90, an ambitious target to help end the AIDS pandemic in 2020: diagnosing 90% of people living with HIV, 90% of diagnosed on treatment, and 90% of people on treatment viral suppressed.”

The inclusion criteria for inclusion of HIV-1-infected individuals in TDR and ADR categories was added in the methods section: “Viral sequences were considered to originate from ART-NP when the first ART regimen was started after the date of resistance testing or at most 6 days prior to the date of resistance testing, to account for delays in the documentation of the date of resistance testing results. Only the first HIV genotypic resistance test per patient was considered for the estimation of TDR for ART-NP. Also, only the first resistance test after the first virologic therapy failure was used for the estimation of ADR in ART-EP.”

In relation to TAMS DRM they were included in NRTIs in all analyses; only in figure 2 (A and B) we separated TAMs from NRTIs.

“Figure 2. Please enlarge the font size for HIV DRM in panel A and B. Explain in the figure legend why 2 histograms are shown for each DRM (panels A and B, except of K65R in the panel A). Why just M184V, K103N, and L90M were selected for presentation in panels C and D?”

Thank you for your comment. The figure has been changed because not all subtitles were visible in Figure 2, but each DRM corresponds to a histogram. We only selected M184V, K103N, and L90M, because we want to analyze the most relevant mutations of each class of antiretrovirals and those mutations that have major importance in the resistance to antiretrovirals used in 1st line therapy.

Minor points

“Line 104: Number after a period (. 67.3…..)”

We have changed this accordingly.

 “lines 119-120: please organize NNRTI and NRTI in the same order in the text as in Table 1”

We have changed this accordingly (lines 137-140).

 “line 151: please indicate a reference for the “previous results”. Please indicate more explicitly that E138A/K mutation confers resistance to NNRTIs (l. 151) and that related calculations concern Table 1.”

This was changed to “Transmitted drug resistance was also calculated considering the NNRTI mutation E138A/K, that confers resistance to Rilpivirine.”

“line 222: strand instead of stand”

We have changed this accordingly.

“line 238: Please move up reference to Table S3 after the first sentence of the paragraph.”

We have changed this accordingly.

 “Please use decimal point in numbers. Sometimes decimal commas are used.”

We have changed this accordingly.

Round 2

Reviewer 1 Report

Dear authors,

many thanks for taking the time to consider my responses - it certainly makes the process feel worthwhile and useful. I agree that the paper has been substantially improved by this process. 

Although I still have some doubts about the p(for trend) analyses, I cannot under the current circumstances refute them sufficiently to continue with the query.

Author Response

To the Board of editors.

We have recently submitted the revision of our manuscript entitled “Increasing prevalence of HIV-1 Transmitted Drug Resistance in Portugal: implications for first line treatment recommendations” to Viruses. We would like to acknowledge all the new comments and suggestions provided by the journal editor and by the reviewers. We are now submitting the second revised version, where we address all the comments and suggestions provided by the reviewers and by the journal editor. We hope to have answered the journal requirements and reviewers’ concerns and to provide now a new final version of the manuscript suitable for publication in Viruses.

Sincerely yours,

Marta Pingarilho

Reviewer 1

Dear authors, many thanks for taking the time to consider my responses - it certainly makes the process feel worthwhile and useful. I agree that the paper has been substantially improved by this process. Although I still have some doubts about the p(for trend) analyses, I cannot under the current circumstances refute them sufficiently to continue with the query.

We would like to acknowledge all the comments provided by reviewer 1 for the previous version of the manuscript. It has led to major improvements of our analyses and we are truly thankful for that. It is certainly worthwhile making science like this.

Reviewer 2 Report

The manuscript have been improved, although I still have some concerns regarding the percentage of ADR and the selection of the patients to estimate ADR. If I understood it correctly, the ART-experienced patients included in this study have had a resistant test done because they have virological treatment failure. Then, why not all patients have DRM? In 2017 of the ART-EP that have had a resistance test only 50.9% have at least one DRM, why the other 49.1% of these patients are not controlling the infection? Is it possible that there are new resistance mutations not included in the Stanford database? Have there been any changes in the criteria for doing a resistance test in patients on ART from 2001 to 2017? Could this influence the drop in DRM observed for this period? On the other hand, selecting a group of treated patients that are not controlling the viremia does not seem a good way to measure ADR. This at most can measure ADR in that specific population, but not overall ADR in Portugal. I might be missing something, but I have the feeling that there is a bias in the selection of the population included in the ART-EP group.

Minor issues:

Figures have been improved; however, I think the resolution and the font size is still low. In addition, there are still some minor English issues (for example, the word “presented” or “presenting” is repeated too many times in the text).

Author Response

To the Board of editors.

We have recently submitted the revision of our manuscript entitled “Increasing prevalence of HIV-1 Transmitted Drug Resistance in Portugal: implications for first line treatment recommendations” to Viruses. We would like to acknowledge all the new comments and suggestions provided by the journal editor and by the reviewers. We are now submitting the second revised version, where we address all the comments and suggestions provided by the reviewers and by the journal editor. We hope to have answered the journal requirements and reviewers’ concerns and to provide now a new final version of the manuscript suitable for publication in Viruses.

Sincerely yours,

Marta Pingarilho

Reviewer 2

“The manuscript have been improved, although I still have some concerns regarding the percentage of ADR and the selection of the patients to estimate ADR. If I understood it correctly, the ART-experienced patients included in this study have had a resistant test done because they have virological treatment failure. Then, why not all patients have DRM? In 2017 of the ART-EP that have had a resistance test only 50.9% have at least one DRM, why the other 49.1% of these patients are not controlling the infection? Is it possible that there are new resistance mutations not included in the Stanford database? Have there been any changes in the criteria for doing a resistance test in patients on ART from 2001 to 2017? Could this influence the drop in DRM observed for this period? On the other hand, selecting a group of treated patients that are not controlling the viremia does not seem a good way to measure ADR. This at most can measure ADR in that specific population, but not overall ADR in Portugal. I might be missing something, but I have the feeling that there is a bias in the selection of the population included in the ART-EP group.

We would like to thank the reviewer for the comment. We agree that it seems counterintuitive that almost half of the population tested for ARV-DR does not have ADR. However, while this value doesn’t surprise us given that our ADR decrease are in line with other studies performed in European countries (Rocheleau G et al., 2018; De Luca A, et al., 2013; de Mendoza C, et al., 2007), we explored the different potential explanations for this:

  • Some patients fail therapy without detectable drug resistance mutations. This could be caused either by previously undescribed mutations (e.g. not included in the Stanford/IAS algorithms), by mutations present in other parts of the HIV genome which were not sequenced or by mutations present in viral minority populations. While we recognize that this is a limitation of our study, we don’t see reasons to believe that the proportion of patients for which we don’t detect DRMs should be changing across time. As such, while our values of ADR could be underestimed due to these facts, the observed trends should not change due to this fact.
  • On the other hand, many patients could perform an ARV-DR test for suspect virological failure and be found instead to have:
    • Bad adherence or treatment abandonment
    • Unexplained low level viraemia
    • Treatment regimen switch due to updated treatment guidelines
    • Loss to follow up

To better clarify this issue, we have changed the methods to reflect that ARV-DR testing was performed in treated patients when patients met the criteria defined by IAS for DR testing, e.g. when:

  • Patients are on antiretroviral treatment and have plasma HIV RNA that is rising to above 200 copies/mL by confirmed measurements after they have been suppressed to below 50 copies/mL;
  • Patients have not achieved full virus suppression after initiating ART
  • Patients have interrupted ART containing an NNRTI with a long half-life (eg, efavirenz);
  • Patients that will switch therapy (e.g. to the use of antiretrovirals with a higher genetic barrier)
  • Patients whose follow-up was lost and return to hospitals to start again ARV-therapy

The sentence in lines 96-98 was changed to “Also, only the first resistance test after the first suspected virological failure was used for the estimation of ADR in ART-EP.”

Minor issues:

Figures have been improved; however, I think the resolution and the font size is still low. In addition, there are still some minor English issues (for example, the word “presented” or “presenting” is repeated too many times in the text).

Following this comment, we have improved this issue.